# Structural basis for allosteric regulation of Human Topoisomerase IIα

Arnaud Vanden Broeck [1,2], Christophe Lotz [1,2], Robert Drillien[1,2], Léa Haas [1,2], Claire Bedez[1,2] & Valérie Lamour [1,2,3✉]

The human type IIA topoisomerases (Top2) are essential enzymes that regulate DNA topology and chromosome organization. The Topo IIα isoform is a prime target for anti-neoplastic compounds used in cancer therapy that form ternary cleavage complexes with the DNA. Despite extensive studies, structural information on this large dimeric assembly is limited to the catalytic domains, hindering the exploration of allosteric mechanism governing the enzyme activities and the contribution of its non-conserved C-terminal domain (CTD). Herein we present cryo-EM structures of the entire human Topo IIα nucleoprotein complex in different conformations solved at subnanometer resolutions (3.6–7.4 Å). Our data unveils the molecular determinants that fine tune the allosteric connections between the ATPase domain and the DNA binding/cleavage domain. Strikingly, the reconstruction of the DNA-binding/cleavage domain uncovers a linker leading to the CTD, which plays a critical role in modulating the enzyme's activities and opens perspective for the analysis of post-translational modifications.

[1] Université de Strasbourg, CNRS, INSERM, Institut de Génétique et de Biologie Moléculaire et Cellulaire (IGBMC), Illkirch, France. [2] Department of Integrated Structural Biology, IGBMC, Illkirch, France. [3] Hôpitaux Universitaires de Strasbourg, Strasbourg, France. ✉email: lamourv@igbmc.fr

Type II DNA topoisomerases (Top2) are evolutionary conserved enzymes whose primordial activity is to regulate the homeostasis of DNA topology in eukaryotes and bacteria[1]. The Top2 are involved in essential cellular processes such as DNA replication, DNA transcription, and chromosome segregation[2]. The human topoisomerase IIα isoform (hTopo IIα) is highly expressed during mitosis, essential for cell division[3] and a biomarker for cell proliferation[4]. As such, Topo IIα is a major target for antineoplastic drugs that hamper its catalytic activities[5].

This large homodimeric enzyme introduces a double-strand break in a first DNA duplex, called G-segment, and directs the transport of a second DNA duplex, called T-segment, through the transient break in order to change the topology of a DNA crossover. The passage of the T-segment requires ATP hydrolysis and is thought to occur along with the opening and closing of several dimeric interfaces constituting molecular gates[6,7]. The crystal structures of the ATPase and DNA binding/cleavage domains of eukaryotic Top2 have been determined and present cavities compatible with the binding of a DNA double helix[8–13]. Biochemical and structural studies have provided evidence that the ATPase domain or N-gate, and the DNA binding/cleavage domain forming the DNA- and C-gates, are allosterically connected, a key feature of its activity[14,15]. However, hinge regions connecting the catalytic sites of the human enzyme remain largely unexplored, hindering efforts to apprehend the quaternary organization of this enzyme and the landscape of conformations it adopts during the catalytic cycle.

In addition, the Top2 catalytic domains are flanked by C-terminal extensions that vary from one species to another[16,17]. These domains contain nuclear localization signals and are submitted to extensive post-translational modifications that condition the cellular localization of Top2, its interactions with cellular partners and progression of the cell cycle[18,19]. Several studies have suggested that different regions of the Topo IIα CTD contribute to the enzyme's catalytic activities through DNA binding[20–23]. In contrast with prokaryotic enzymes that harbor a pinwheel-structured CTD[24–26], the same region in eukaryotic enzymes presents no homology to any known fold, hence limiting structure–function analysis. It has become clear that the analysis of the molecular determinants of the enzyme's allostery and the modulation of its activity by the CTD now requires the availability of a complete molecular structure of the Topo IIα.

In this work, we determined the cryo-EM structure of the full-length human Topo IIα isoform bound to DNA in different conformations trapped by the anti-cancer drug etoposide. The structures reveal the connections between the ATPase and DNA binding/cleavage domains, allowing the identification of conserved sequence patterns in humans that control the allosteric signaling between the catalytic sites. In addition, we were able to localize the linker between the DNA binding/cleavage domain and the CTD inserting below the G-segment. We show that this region directly stimulates the Topo IIα catalytic activity suggesting that the bulk of the CTD domain may counterbalance this effect, potentially through post-translational modifications.

## Results

**Cryo-EM reconstructions of the hTopo IIα nucleoprotein complex.** The full-length hTopo IIα was overexpressed in mammalian cells and purified using tandem affinity chromatography followed by a heparin step (see the "Methods" section and Supplementary Fig. 1). Samples were tested for enzymatic activity and mixed with asymmetric DNA oligonucleotides mimicking a doubly nicked 30 bp DNA[11] (Supplementary Fig. 1b and Table 2). The antineoplastic drug etoposide and AMP-PNP, the non-hydrolysable homolog of ATP, were added to the nucleoprotein complex in order to minimize the conformational heterogeneity of the sample.

The DNA-bound hTopo IIα complex was analyzed by single-particle cryo-EM. Extensive data collection and ab-initio 3D classifications and focused refinement strategies enabled us to solve structures of the entire human Topo IIα in different conformations. The flexibility of the complex is visible from the 2D class averages showing the dimerized ATPase domain can adopt different positions relative to the DNA-binding/cleavage domain (Supplementary Figs. 1e and 3). A first consensus structure was solved at 6.6 Å (Supplementary Fig. 3a). To isolate the different conformations and improve the overall resolution, we used a combination of global and local approaches to deconvolute the structures of the hTopo IIα complex. Using 3D focused refinements, reconstructions of the DNA-binding/cleavage domain were generated for two different states. State 1 was solved at 3.6 Å using 60,601 particles and State 2 was solved at 4.1 Å using 39,368 particles (Supplementary Fig. 3d, f). Each particle stack was submitted to 3D heterogeneous classification and refinement without mask to yield two reconstructions of the entire complex in State 1 at 4.7 Å resolution from 26,506 particles, and in State 2 at 7.4 Å resolution from 13,420 particles (Supplementary Fig. 3e, g). Due to the flexibility of the ATPase domain, the densities of the linkers with the DNA binding/cleavage domain were not well resolved in the cryo-EM maps (Fig. 1c). To get information on this region, a 3D-focused classification was performed on the ATPase domain yielding one class comprising 36,610 particles with a well-resolved connection between the two functional domains which was refined at 7.6 Å resolution (Supplementary Fig. 3b).

**Model building of the hTopo IIα complex in different states.** The well-resolved EM density of the 3.6 Å map in State 1 allowed us to fit, build and refine the atomic model of the hTopo IIα DNA-binding/cleavage domain in complex with DNA and etoposide[27] (Fig. 1, Supplementary Fig. 5). Strikingly, we were also able to identify EM density for the etoposide molecule, intercalating in the DNA duplex between positions −1 and +1 (Fig. 1e), with similar protein and DNA contacts as previously reported in crystallographic structures (Supplementary Fig. 7)[10,27].

The resulting DNA-binding/cleavage domain model together with the crystal structure of the ATPase domain bound to AMP-PNP[8] were then combined to refine the complete atomic structure of the full-length hTopo IIα in State 1 (Fig. 2 and Supplementary Movie 1). The 27-aa linkers between the two domains, missing from the crystal structure of the isolated ATPase domain, were built as alpha helices based on secondary structure predictions and using well-defined linker densities of the cryo-EM map obtained by focused classification on the ATPase domain (Supplementary Figs. 3b and 6). Finally, the atomic model of the DNA-binding/cleavage domain in State 2 and the corresponding full-length hTopo IIα in State 2 were refined using State 1 model as a reference.

**Analysis of the hTopo IIα conformations.** The complete architecture of the hTopo IIα reveals the intertwined arrangement of the two subunits, a feature that was deduced from the yeast enzyme crystal structure and recently observed in the cryo-EM models of the prokaryotic Topo II[12,28] (Figs. 2d and 3). The dimeric ATPase domain sits in a ~95° orthogonal orientation above the DNA-gate, similar to what was previously observed with the yeast Topo II (~90° orientation)[12] (Fig. 3e–g). The structure is asymmetric with the ATPase domain slightly bent (~10°) in both orthogonal plans (Figs. 2c and 3d, e). Compared to the yeast Topo II structure, the ATPase domain is positioned 15 Å

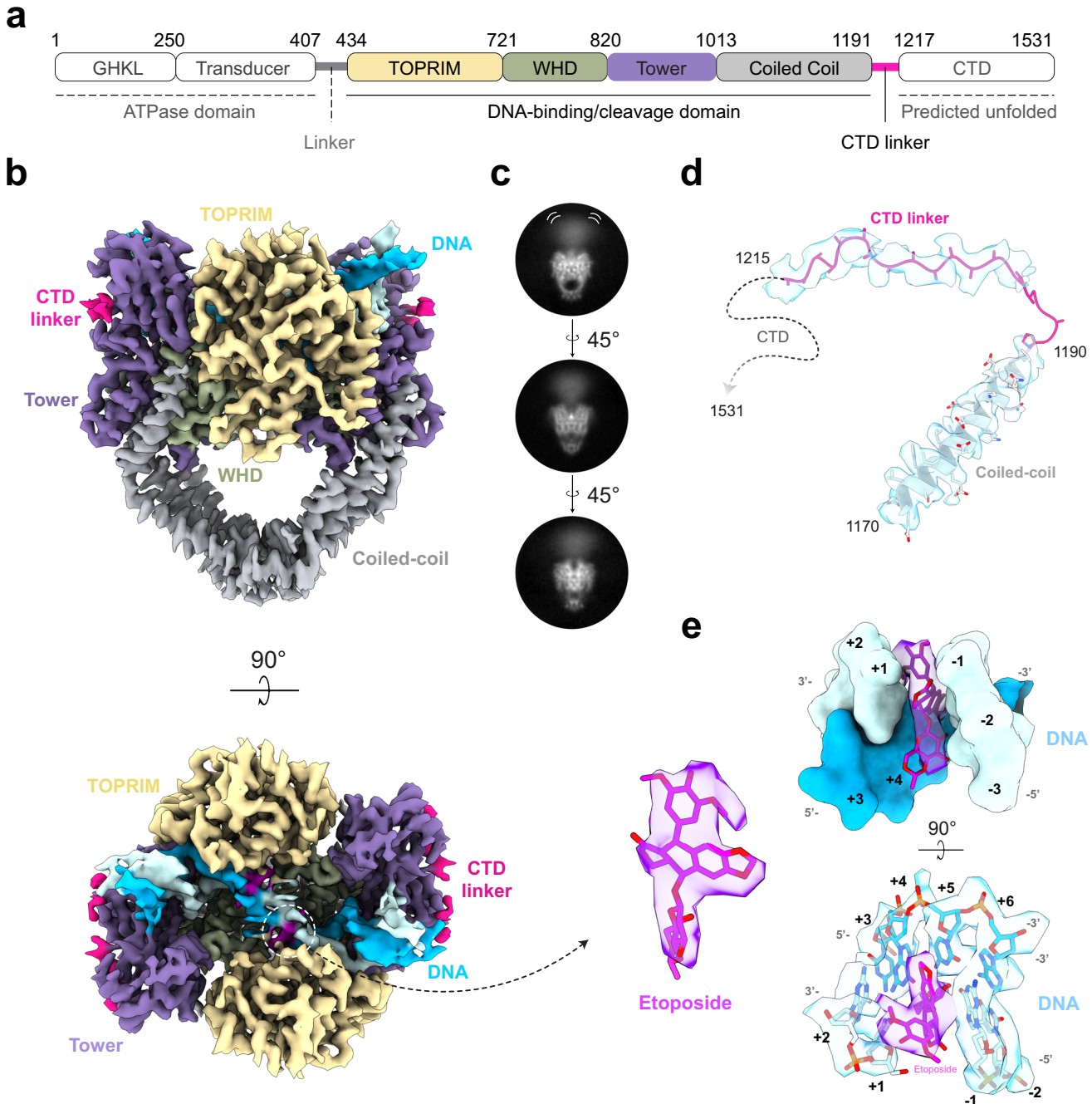

**Fig. 1 Cryo-EM structure of the hTopo IIα etoposide-poisoned DNA-binding/cleavage domain. a** Schematic representation of the hTopo IIα DNA-binding/cleavage domain. **b** Cryo-EM structure of the DNA-binding/cleavage domain in State 1 solved at 3.6 Å resolution. The structure is colored as in **a**, the DNA is colored in blue. **c** 2D classes of hTopo IIα in different orientations showing the flexibility of the ATPase domain. **d** EM density around the last coiled-coil helix and the CTD linker. **e** Zoom on the EM density of the etoposide and its binding site intercalating DNA bases +1 and −1.

above the DNA-gate. Interestingly this creates a cavity large enough to accommodate a T-segment on top of a G-segment (Fig. 3a, e, f).

In State 1, the TOPRIM and Tower domains are positioned upward and tightly bound to the G-segment which is highly bent. This state corresponds to a closed cleavage complex as observed in the structure of the DNA-bound hTopo IIα DNA-binding/cleavage domain crystallized without etoposide, with the exception of the C-gate being closed in our structure[11]. In State 2, the TOPRIM domain is moved upward by 7 Å while the Tower domain is moved away from the TOPRIM domain by 8 Å (Fig. 3b). The physical separation of the TOPRIM and tower

domains induces the stretching and unwinding of the G-segment by 8 Å in both directions (Fig. 3d), positioning the DNA-binding/cleavage domain in a pre-open conformation as observed in the crystal structure of DNA/etoposide-bound hTopo IIα DNA-binding/cleavage domain[27]. In this conformation, the TOPRIM and tower domains are separated by ~20 Å, which could contribute to the formation of the cavity compatible with the presence of a T-segment before transport through the G-segment (Fig. 3a).

Since the two structures solved in different states are bound to AMP-PNP and etoposide, the complexes are trapped in a form precluding the opening of the G-segment. However, the hTopo

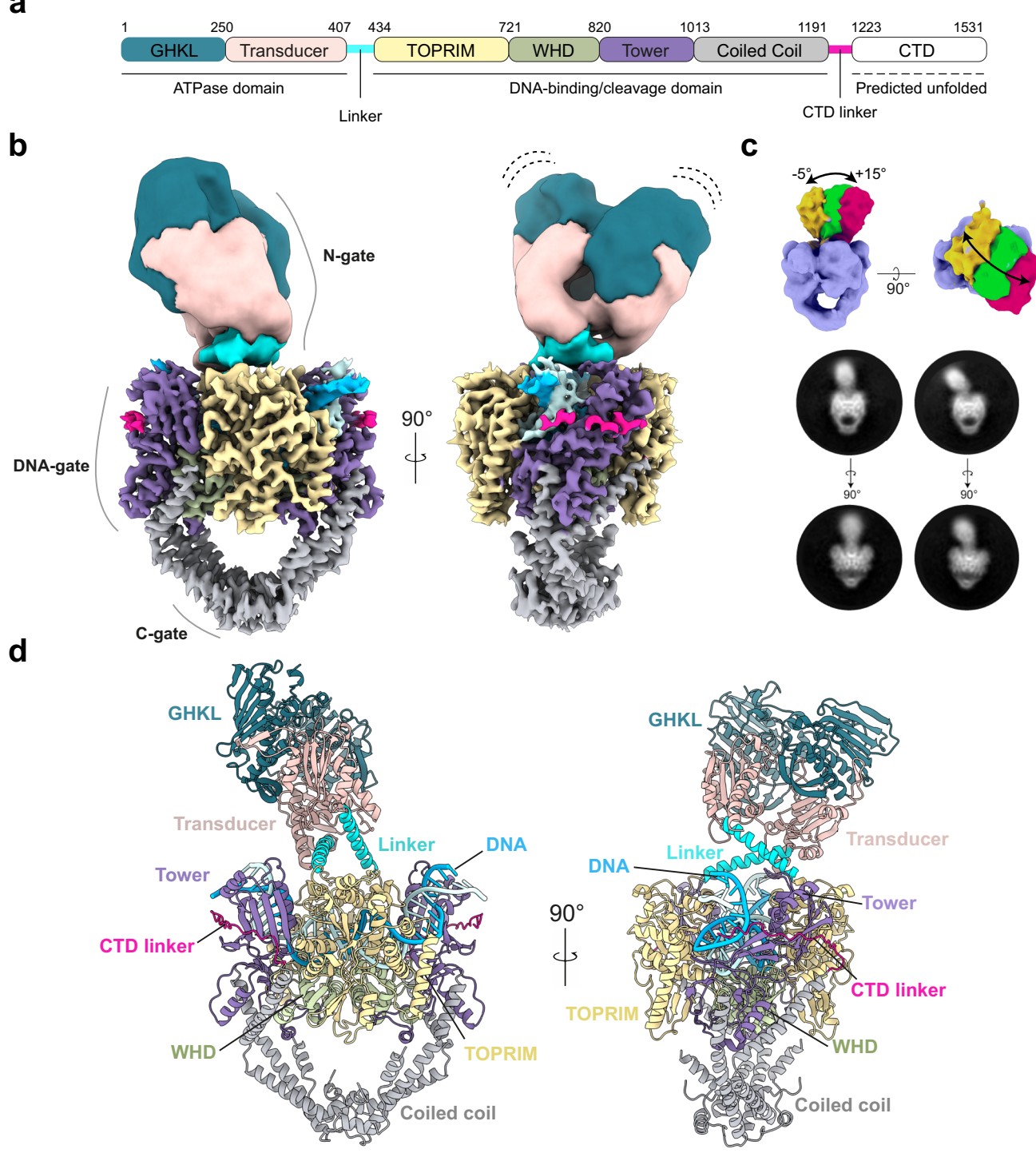

**Fig. 2 Molecular model of the DNA-bound hTopo IIα. a** Schematic representation of the hTopo IIα domains. Human Topo IIα is composed of two identical subunits assembling in an active homodimeric form. **b** Composite cryo-EM map of the full Human Topo IIα with DNA-gate in closed state. **c** 3D and 2D classes showing the flexibility and motions of the ATPase domain with respect to the DNA-binding/cleavage domain. **d** Atomic model of the full-length Human Topo IIα. The structure is colored as in **a**, the DNA is colored in blue.

IIα DNA binding/cleavage domain is still able to oscillate between the closed and pre-open states in presence of etoposide, despite the fact that the G-segment base pairs remain annealed (see supplemental analysis in the Supplementary information). This conformational oscillation has also been observed in different bacterial DNA gyrase complexes bound to antibiotics[28,29] (Fig. 3, Supplementary Figs. 3 and 8).

We also found that the ATPase domain adopts different inclinations depending on the conformation of the DNA-binding/cleavage domain. When the DNA-binding/cleavage domain is a closed state, the ATPase domain is bending over by 5° relative to the dimeric symmetry axis. In the pre-open state, the ATPase domain can bend over by about 15° (Fig. 2c Supplementary Fig. 3d). These tilting movements are inherent to the flexibility of the ATPase domain which sits on top of a cavity

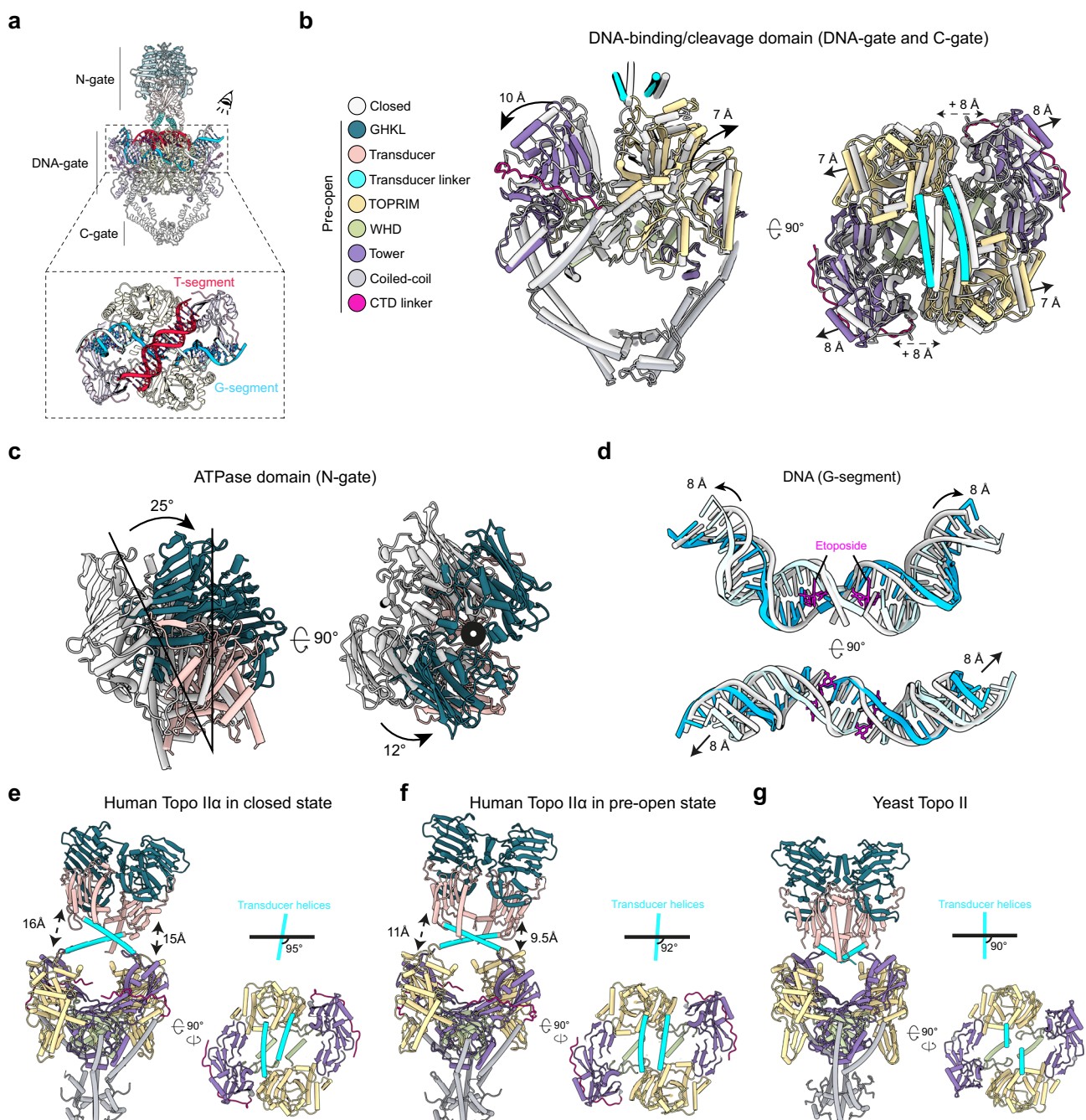

**Fig. 3 Structural changes associated with G-segment opening and allosteric connections between the N- and DNA-gates. a** Schematic of the different modules of hTopo IIα and DNA segments. The T-segment in red is modeled. **b** Superimposition of DNA-binding/cleavage domain structures in State 1 (closed) (gray) and State 2 (pre-open) (colored based on the legend). TOPRIM domain and Tower domain are moving away from 8 Å in the pre-open state. **c** A 25° lateral movement of the ATPase domain together with a 12° rotating movement of the ATPase domain (opposite to the intertwining) is observed during the transition from closed to pre-open conformation of the DNA-gate. **d** During the transition, the DNA is unwound and stretched by 8 Å. **e** hTopo IIα with DNA-gate in closed state: the ATPase domain is bent by ~5° with respect to the vertical axis and also by ~25° in the orthogonal plane, positioning the ATPase domain at a distance of ~15 Å of the DNA-gate (left). The ATPase domain transducer alpha-helices in cyan form a ~95° angle with the DNA-gate (right). **f** hTopo IIα with DNA-gate in pre-open state: the ATPase domain is bent by ~8° with respect to the vertical axis and also by ~5° in the orthogonal plan, positioning the ATPase domain at a distance of ~10 Å of the DNA-gate (left). The ATPase domain transducer alpha-helices in cyan form a ~92° with the DNA-gate (right). **g** Yeast Topo II with DNA-gate in closed state[12]: the ATPase domain is straight in both orthogonal plans, positioning the ATPase domain in close contact with the DNA-gate (left). The ATPase domain transducer alpha-helices in red form a ~90° with the DNA-gate (right).

only partially occupied by the DNA G-segment. It is most likely that the presence of a fully formed DNA crossover would constrain the tilting range of the N-gate.

The conformational changes observed in the DNA-gate are however directly correlated to rotations and translational movements of the ATPase domain (Fig. 3b, c). Conformational changes of the TOPRIM domains in the DNA-gate modify the distance between residues N433 of each monomer, which are connected to the alpha helices of the ATPase transducers. In the closed state, the distance between the two residues N433 is 49.3 Å while the distance is shortened to 46.4 Å in the pre-open state. Consequently, this translational movement of 3 Å in the orthogonal plane of the DNA gate induces the rotation of the ATPase domain by 12° counterclockwise, opposite to the intertwining (Fig. 3c). It also forces the ATPase domain to come closer to the DNA-gate prefiguring a conformation that would position the T-segment in the newly formed groove between the TOPRIM and tower domains (Fig. 3f). Altogether the rotation of the N-gate that correlates with the oscillation of the DNA-gate can be associated with a corkscrew mechanism that we can extrapolate from the two overall conformations of the full-length catalytic core (Supplementary Movie 2).

**Molecular determinants of allosteric transitions**. Although the N-gate is not required for G-segment cleavage, the DNA gate per se is not able to open unless ATP binds to the N-gate[30]. These findings support a model that implies a direct coupling between the ATP binding/hydrolysis and the DNA-gate opening through the 27-aa alpha helices linkers.

To analyze the molecular determinants of this allosteric mechanism, we performed a sequence analysis of the TOP2 protein from 30 species of metazoan and multicellular plant species, including Topo IIα and Topo IIβ from five vertebrates (Fig. 4a, b). The conservation profile of the 27-aa linker, predicted to fold as an alpha helix, revealed four highly conserved residues among the superior eukaryotes: $W_{414}$, $F_{417}$, $K_{418}$, and $K_{425}$ (Fig. 4b). The two aromatic residues $W_{414}$ and $F_{417}$ form a hydrophobic patch between the linkers, which could contribute to the stability of their interaction (Fig. 4d). Lysine$_{418}$ is close to the K-loop (342–344), that was shown to be involved in DNA sensing in the yeast enzyme[12]. Residue $K_{425}$ is also highly conserved and is located towards the end of the linker helices, at the entrance of the TOPRIM domain (Fig. 4d). To assess the contribution of these residues in the allosteric regulation of the human enzyme, we designed four hTopo IIα mutants: $K_{418}A$ to remove the positively charged side chain close to the K-loop, $K_{425}A$ at the end of the transducer helices, $W_{414}A$-$F_{417}S$ to remove the hydrophobic patch and $K_{425}G$-$K_{426}G$-$C_{427}P$ to disrupt the alpha helix fold. We tested their ability to perform DNA relaxation and DNA cleavage and their ATPase activity in comparison with the wild-type enzyme.

Despite affecting highly conserved residues, the $K_{418}A$ and $K_{425}A$ single-mutant proteins display DNA relaxation, cleavage, and ATPase activities in the same range as the WT enzyme (Fig. 4c, e, f and Supplementary Fig. 9a, b, e). These two residues alone do not seem to predominantly affect the interaction networks around the transducer during the catalytic cycle. The $W_{414}A$-$F_{417}S$ double mutant shows only a slightly decreased relaxation and ATPase activities, but a 5-fold decrease in cleavage activity in presence of etoposide (Fig. 4c, e, f, Supplementary Fig. 9b, e). Without the presence of the hydrophobic patch, the cleavage complex in presence of etoposide seems less stable than for the WT.

Introduction of the triple mutations $K_{425}G$-$K_{426}G$-$C_{427}P$ shows a 5-fold reduced cleavage activity in presence of etoposide,

similarly to the double mutant $W_{414}A$-$F_{417}S$. However, the relaxation activity is impaired compared to the WT enzyme and the ATPase activity is reduced 2-fold (Fig. 4c, e and Supplementary Fig. 9a, b, e). Introduction of the mutations in the middle of the helices are likely to disrupt the helical structure, loosen the linkers and decrease their stability. The ATPase activity of the Top2 has been shown to be stimulated by DNA binding[31]. ATPase assays performed in absence of DNA show that most single and double mutants show a 20% decreased ATPase activity as for the WT. Albeit starting from a lower level of activity this decrease is stronger for the triple mutant (about 50%), showing that this region specifically affect allosteric communication between the DNA-binding/cleavage domain and ATP hydrolysis.

**Regulation of the catalytic activity by the C-terminal domain**. Although we used the complete hTopo IIα sequence for our structural analysis by cryo-EM, we were not able to observe any density belonging to the bulk of the CTD in the 2D classes, nor in the 3D reconstructions. Secondary structure prediction on the CTD sequence (residues 1191–1531) suggests that this domain is disordered (Fig. 5a). We analyzed the CTD structure in solution using small angle X-ray scattering (SAXS). The Kratky plot derived from the scattering curve showed a plateau for high scattering vector $q$-values, typical of unfolded proteins, in contrast with the Gaussian curve observed in globular domains[32] (Supplementary Fig. 10). Although complementary experiments would be required to conclude on the CTD structure, it suggests that in these conditions this domain is disordered or highly flexible. However, analysis of the 3.6 Å EM map of the DNA binding/cleavage domain revealed an additional EM density that could be attributed to the beginning of the CTD domain (residues 1192–1215) (Figs. 1d and 5c and Supplementary Fig. 5g). This region of the CTD begins at the end of the terminal coiled-coil alpha helix of the DNA binding/cleavage domain on residue 1192, stretches along the Tower domain and extends under the G-segment major groove pointing in an orthogonal direction from the DNA gate (Fig. 5c).

To our knowledge, the linker leading to the CTD has not been previously observed in a eukaryotic Top2 structure. The CTD has been shown to confer specific functions and DNA topology preferences to the human isoforms that differ in this region[33,34]. It also undergoes multiple post-translational modifications that regulate its cellular distribution and activity throughout the cell cycle[2]. Several studies have explored how the CTD could modulate the catalytic activities and DNA-binding properties of the Top2[21,35]. Although previous studies have examined the effect of CTD deletion at different positions, the Top2 constructs were ending before, or at position 1192, therefore not accounting for the contribution of this linker region that is in close proximity with the G-segment[36]. To assess the contribution of this region in the catalytic activities, we designed hTopo IIα constructs with a complete deletion of the CTD (Δ1193) or a partial CTD truncation (Δ1217) (Fig. 5b).

The hTopo IIα lacking the complete CTD (Δ1193) showed similar relaxation activity as the WT enzyme, as also observed in previous studies[36] and ATPase activity slightly lower than the WT Topo IIα within error margin (Fig. 5d, e, and Supplementary Fig. 9). The mutant Δ1193 is however impaired in its cleavage activity in presence of etoposide compared to the WT (Fig. 5f and Supplementary Fig. 9). A similar effect of a larger Δ1175 CTD deletion was already observed with high concentrations of drug, showing that loss of the complete CTD decreases etoposide cleavage[35]. On the contrary, the relaxation activity of the hTopo IIα with the partial CTD truncation (Δ1217) was increased ~10-fold compared to the wild-type or Δ1193 enzymes, as well as the

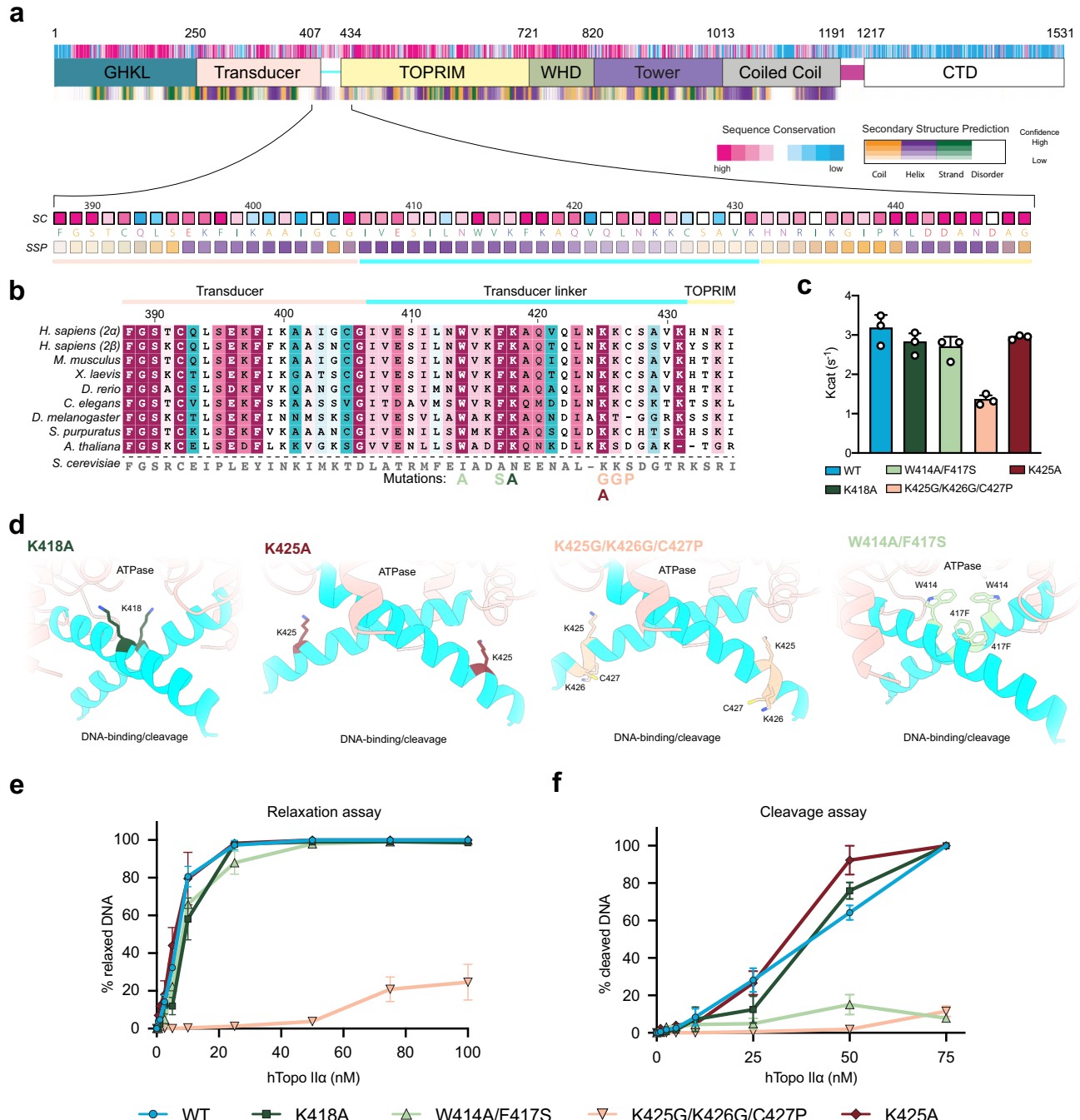

**Fig. 4 Analysis of the hTopo IIα allosteric regulation mediated by the linker connecting the N-gate to the DNA-gate. a** Domain organization, sequence conservation, and secondary structure prediction for hTopo IIα. Below is a zoom view of the linker joining N-gate to DNA-gate showing the sequence at a residue level. SC stands for sequence conservation. SSP stands for secondary structure prediction. **b** Multiple sequence alignment focused on the linker region. The yeast sequence is displayed as information but was not included in the conservation calculation. **c** ATP hydrolysis activity of the wild-type, $K_{418}A$, $W_{414}A$-$F_{417}S$ and $K_{425}G$-$K_{426}G$-$C_{427}P$ and $K_{425}$ hTopo IIα. Kcat values are presented as mean values ± standard error (SE) for three independent replicates ($n = 3$). Individual data points are also plotted. **d** Spatial localization of the mutated residues on the linker joining N-gate to DNA-gate. **e** Relaxation and cleavage **f** activities for the wild-type, $K_{418}A$, $W_{414}A$-$F_{417}S$, $K_{425}G$-$K_{426}G$-$C_{427}P$ and $K_{425}A$ hTopo IIα. Zoomed panels on the region 0–10 nM is available in Supplementary Fig. 9. Data are presented as mean values ± SE for three independent replicates ($n = 3$). The source data for panels **c**, **e**, and **f** are provided as a Source Data file.

cleavage activity (Fig. 5e, f and Supplementary Fig. 9c, d). At the same time, the Δ1217 mutant maintains its ATPase activity, slightly higher than the WT enzyme within error margin (Fig. 5d and Supplementary Fig. 9e).

The portion of the CTD ranging from residue 1192 to residue 1217 is in close vicinity of the G-segment and seems to act as a positive regulator of the strand passage process, which appears to stimulate the relaxation activity and stabilize the cleavage complex in presence of etoposide. Remarkably, cleavage assays in absence of ATP show that the Δ1217 mutant is still able to maintain a high rate of cleavage in presence of etoposide in contrast with the WT and the Δ1193 proteins (Supplementary

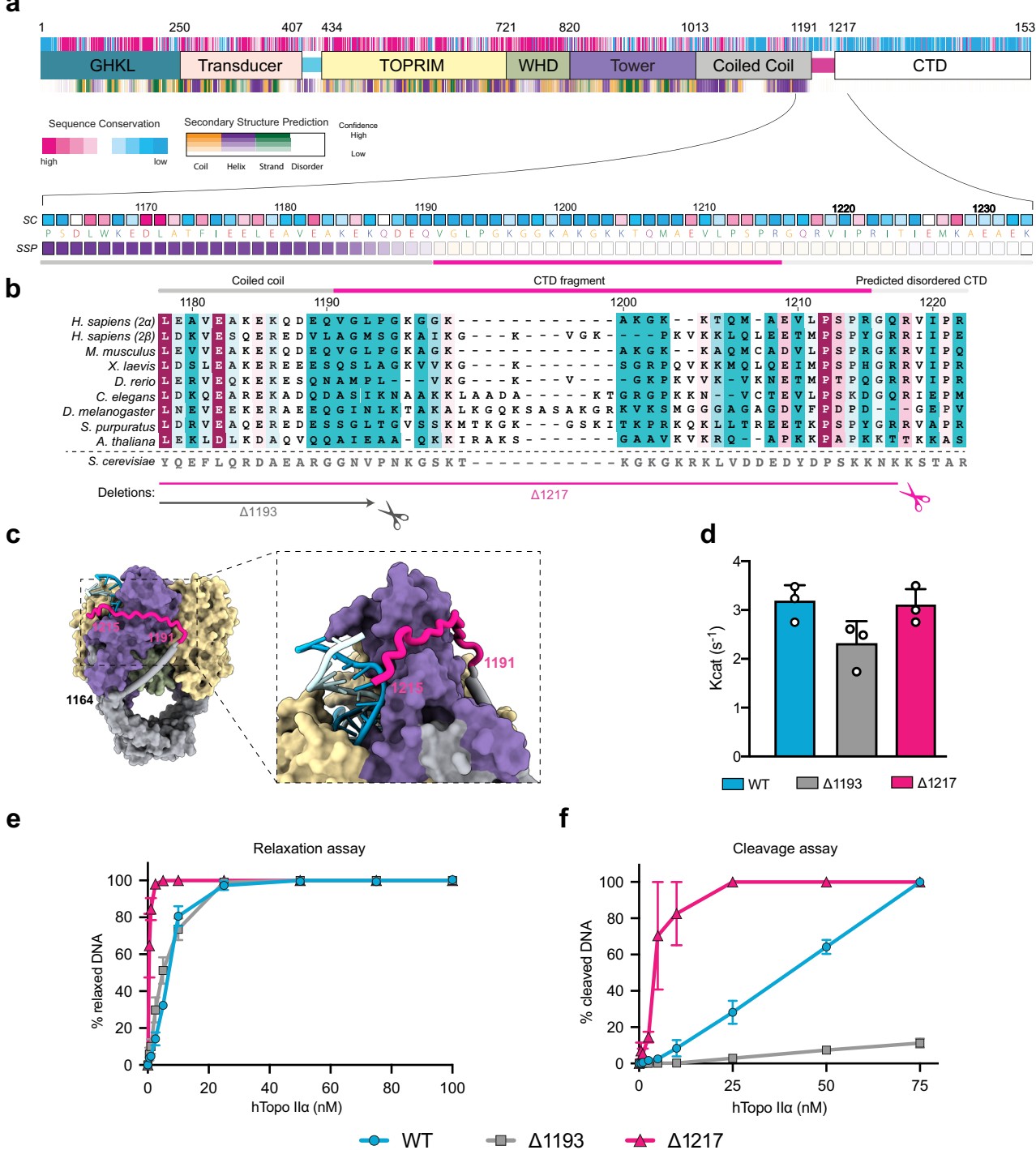

**Fig. 5 Regulation of the catalytic activity by the C-terminal domain linker. a** Domain organization, sequence conservation, and secondary structure prediction for hTopo IIα. Below is a zoom view of the CTD fragment visible in the EM reconstruction showing the sequence at a residue level. SC stands for sequence conservation. SSP stands for secondary structure prediction. **b** Multiple sequence alignment focused on the CTD linker region. The yeast sequence is displayed as information but was not included in the conservation calculation. The CTD deletions are shown below. **c** Orthogonal views of the DNA-binding/cleavage domain in closed state. The CTD domain is highlighted in pink. The inset shows a zoom on the region of the CTD which comes in close vicinity of the DNA G-segment. **d** ATP hydrolysis activity by wild-type, Δ1193 and Δ1217 hTopo IIα. Kcat values are presented as mean values ± SE for three independent replicates (n = 3). Individual data points are also plotted. **e** Relaxation and cleavage **f** activities for the wild-type, Δ1193 and Δ1217 hTopo IIα. Zoomed panels on the region 0–10 nM is available in Supplementary Fig. 9. Data are presented as mean values ± SE for three independent replicates (n = 3). The source data for panels **d**–**f** are provided as a Source Data file.

Fig. 9d). Binding of ATP closes the N-gate dimeric interface on top of the DNA-binding cavity. This structural feature is important for the stabilization of the cleavage complex in presence of etoposide, as shown by impaired activities in presence of mutations in the transducer helices. The Δ1217 mutant, despite of the absence of a CTD but keeping the 1193–1217 linker is however able to overcome the lack of dimerization, indicating a key role of this linker in allosteric signaling.

## Discussion

The cryo-EM structures of the entire hTopo IIα reveal how the ATPase domain is spatially connected to the DNA-binding/cleavage domain conformations. The alpha helices that were missing in crystal structures of the human ATPase domain, are reminiscent of the transducer helices found in prokaryotic Top2 enzymes but differ in sequence and in the relative orientation of their crossing, which narrows down the cavity of the N-gate (Figs. 3, 4 and Supplementary Fig. 6). Previous crystal structures of type 2 Top2 isolated domains or cryo-EM reconstructions of the bacterial homolog DNA gyrase reported distinct DNA/binding-ing domain conformation but so far without connection with the position of the ATPase domain[15,28,37]. Our analysis of two highly conserved lysine residues along the transducer helices shows that single positions do not seem to play a role in the allosteric communications of the Top2 catalytic core. However, mutations disrupting the $W_{414}$-$F_{417}$ hydrophobic patch only show minor decrease in DNA relaxation rate and ATP hydrolysis but a major decrease in cleavage activity suggesting that interactions locking together the two transducer linker helices contribute to the stability of the etoposide binding site. Finally, introduction of triple mutation that destabilize the helical structure in the transducer is affecting not only the cleavage complex as for the double mutant $W_{414}$A-$F_{417}$S but also transmission of the allosteric signal from the ATPase domain. All together our data suggest the transducer helices structure and specific motifs per se represent an important feature for the regulation of allosteric movements. Our structures highlight how subtle movements in the DNA-gate propagate to the N-gate and vice versa through networks of interactions mediated by conserved motifs, in particular in the transducer helices.

Although AMP-PNP, a non-hydrolysable analog of ATP, was used to dimerize the ATPase domains, this shows that conformational transitions can occur within the same sample. Indeed, it was shown that AMP-PNP can support a single, complete round of DNA passage, but that the ATPase domains remain dimerized, preventing further rounds of activity[38,39]. Structures with other nucleotides would provide further insights in the conformational range that link rotation of the N-gate to opening of the DNA gate. Although the isolated N-gate of prokaryotic enzymes have been shown to bind a DNA duplex within its cavity[40], it is unlikely that the human enzyme could accommodate a DNA double helix without subsequent conformational rearrangement and partial opening of the transducer helices.

In addition, the 3.6 Å structure of the DNA-binding/cleavage domain reveals a part of the CTD domain which was not previously observed and is positioned nearby the G-segment inflection points on each side of the enzyme. The curvature of the DNA was shown to be important for the selection of cleavage sites[41]. The particular localization of the CTD linker may structurally favor the curvature of the G-segment, stimulating DNA cleavage and favoring strand passage. We also show that this linker preceding the CTD domain enhances the catalytic activities of the hTopo IIα. As the Δ1193 mutant devoid of this linker and of the CTD shows a similar relaxation activity as the full-length hTopo

IIα, this suggests that the bulk of the CTD domain would counterbalance the stimulating effect of the linker.

Interestingly the orientation of the linkers suggests the CTD domain could be positioned in an orthogonal direction relative to the plane of the Top2 catalytic domains. On the contrary, in prokaryotic enzymes, the linkers of the CTD domains are shorter and positioned in the same plane (Supplementary Fig. 5)[24,28]. The CTD linker sequence of the Topo IIα displays a high content in positively charged residues, as well as several conserved residues at positions 1209 and 1212–1214 also found in the Topo IIß isoform and other eukaryotic species (Fig. 5b). This suggest that the CTD linker constitutes a common structural feature of the multicellular eukaryotic Top2, independently of the conservation of the rest of their CTD.

It is worth noting that $S_{1213}$, located at the end of the linker, has been found to be subjected to mitotic phosphorylation and contributes to localization of the Topo IIα to the centromere[42–44]. Such post-translational modification could regulate the binding of this CTD portion to the G-segment in order to modulate the relaxation activity of the hTopo IIα activities during the cell cycle. The Top2 activities are indeed associated with cellular complexes regulating the genome organization[45]. The chromatin tether, a specific sequence within the hTopo IIα CTD, was shown to interact with histone tails in chromatin structures[18]. The path of the CTD linker along the catalytic core of the enzyme indicates that the CTD may be positioned in a different orientation compared with the prokaryotic homologs, and may favor its binding to nucleosome structures in the eukaryotic genome.

## Methods

**Wild-type hTopo IIα expression and purification.** The sequence coding for the full-length human Topo IIα (1–1531) was inserted into a modified pVote0GW vector depleted of attB1 sequence and containing an N-terminal Twin-strep tag and a C-terminal 10 His-tag. The E. coli gene encoding xanthine-guanine phosphoribosyl transferase (GPT) inserted into a portion of the viral hemagglutinin gene sequence (HA) present in the plasmid was used to select recombinant MVA-T7 viruses holding the hTopo IIα construct (TOP2A gene) under the dependence of a T7 promoter. Resistance to mycophenolic acid (MPA), provided by the GPT gene, enabled selection of recombinant viruses which were subsequently amplified in the absence of MPA[46]. Prior to overexpression, 600 ml of BHK21 C13-2P cells in suspension ($10^6$ cells/ml) were infected with ~0.1 PFU/cell of recombinant virus in cell culture medium (GMEM, 10% FCS, 1.5 g/l BTP, 50 μM Gentamycin) and incubated at 37 °C. After 48 h, the infected cells were mixed with 6 l of uninfected cells at $10^6$ cells/ml and a 1:10 ratio (v/v), respectively. Overexpression was directly induced by the addition of 1 mM IPTG followed by an incubation of 24 h at 37 °C. Cells were harvested and resuspended in Lysis buffer (25 mM Hepes, 500 mM NaCl, 500 mM KCl, 1 mM MgCl₂, 20 mM imidazole, 10% v/v glycerol, 2.5 mM beta-mercaptoethanol, 0.5 mM PMSF, 0.5 mM Pefabloc, protease inhibitor cocktail (Roche), pH 8.0) and lysed with three cycles of high-pressure disruption using an EmulsiFlex-C3 apparatus (Avestin) at 1500 bar. The full-length hTopo IIα was first purified by a tandem affinity chromatography on a manually packed XK 26/20 column (Pharmacia) with Chelating Sepharose 6 Fast Flow resin (Cytiva) bound to Ni²⁺ ions followed by a StrepTrap HP column (Cytiva). Elution from the chelating resin was performed using 250 mM imidazole pH 8.0 added to the Lysis buffer and allowed the protein to bind to the StrepTactin Sepharose resin. The protein was washed with 25 mM Hepes, 200 mM NaCl, 200 mM KCl, 1 mM MgCl₂, 10% v/v glycerol, 2 mM DTT, pH 8.0 and eluted with the same buffer supplemented with 3 mM Desthiobiotin (Sigma). Twin-strep and His tags were removed by the addition of P3C (Precission protease) at 1:50 ratio (w/w) and incubated overnight at 4 °C. The cleaved protein was then loaded on a HiTrap Heparin HP column (Cytiva). Elution was performed by a single step using 25 mM Hepes, 400 mM NaCl, 400 mM KCl, 1 mM MgCl₂, 10% v/v glycerol, 2 mM DTT, pH 8.0. After the purification process (Supplementary Fig. 1a), 20 mg of the full-length hTopo IIα were obtained from 6 l of BHK21 C13-2P cell cultures. About 10–15% of the protein sample shows some degradation depending on the protein batch, as previously observed[42]. Western blot analysis using monoclonal TOP2A antibody 1E2 at dilution 1:1000 (catalog number WH0007153M1—Sigma-Aldrich) shows that the C-terminal domain tends to be cleaved off during protein purification despite the use of protease inhibitors (Supplementary Fig. 1a). However, the majority of the sample is constituted by full-length protein forming an intact homodimer that can be stabilized to form complexes with DNA prior to cryo-EM sample freezing (Supplementary Fig. 1c). Uncropped gels and blots are provided in Source Data.

**Nucleic acid preparation**. A doubly nicked 30 bp DNA duplex was reconstituted using two asymmetric synthetic oligonucleotides obtained from Sigma-Aldrich. Sequences for the single strand 17 bp (5′-CGCGCATCGTCATCCTC-3′) and the single strand 13 bp (5′-GAGGATGACGATG-3′) were chosen as described in ref. [11]. Briefly, the nucleic acids were dissolved in DNAse-free water at 1 mM concentration. To assemble the double-stranded DNA, each oligo was mixed at 1:1 molar ratio, annealed by incubating at 95 °C for 2 min and then decreasing the temperature by 1 °C every 1 min until reaching 20 °C. The annealed doubly nicked DNA duplex was then buffer-exchanged in Hepes 20 mM pH 7.5 with a BioSpin 6 column (BioRad).

**Complex formation for cryo-EM**. Purified hTopo IIα was mixed with the 30 bp dsDNA at 1:1 molar ratio to obtain a 20 μM final concentration of protein and DNA. The mixture was incubated for 10 min at 37 °C. Then, etoposide (Sigma-Aldrich) was added to reach a final concentration of 0.5 mM. The subsequent mixture was incubated for 10 min at 37 °C. Finally, ADPNP was added to the complex at a final concentration of 0.5 mM. The fully reconstituted complex was further incubated 30 min at 30 °C.

**BS3 cross-linked hTopo IIα–DNA–etoposide–ADPNP complex**. The purified hTopo IIα is unstable in buffer conditions under low salt conditions. Therefore, a chemical stabilization by BS3 was necessary for further cryo-electron microscopy analysis. To determine the optimal concentration of BS3 allowing a complete stabilization of the complex without inducing aggregates, the newly formed complex was incubated with different concentration of BS3 and analyzed by SDS–PAGE. Briefly, BS3 (Thermo Fischer Scientific) was freshly resuspended in filtered DNAse-free water at 25 mM stock concentration. Rapidly after BS3 preparation, the complex was incubated for 30 min at 30 °C with 0.25 mM up to 5 mM BS3 and the crosslinking was quenched by adding Tris–HCl, pH 7.5, to 50 mM. The optimal concentration was determined at 1 mM (Supplementary Fig. 1c). The cross-linked complex was centrifuged at $20,000 \times g$ for 30 min at 4 °C to remove remaining aggregates.

**Buffer exchange of the complex in optimal cryo-EM buffer**. The complex was first dialyzed against 200 ml of intermediate buffer (20 mM Hepes, 200 mM KAc, 200 mM Na-Glu, 5 mM MgAc₂, 0.5 mM TCEP, pH 7.5) for 2 h at 4 °C using Slide-A-Lyzer MINI Dialysis Units (7000MWCO) (Thermo Fischer Scientific). Then, a second dialysis was performed in 200 ml of the final cryo-EM buffer (20 mM Hepes, 100 mM KAc, 50 mM Na-glutamate, 5 mM MgAc₂, 0.5 mM TCEP, pH 7.5) for 4 h at 4 °C. Finally, 8 mM CHAPSO (Sigma-Aldrich) was added to the dialyzed complex to prevent adsorption of the particles to the air/water interface[47]. The sample was centrifuged for 2 h at $16,000 \times g$ to remove potential aggregates.

**Cryo-EM grid preparation**. Quantifoil R-1.2/1.3 300 mesh copper grids were glow-charged for 20 s prior to the application of 4 μl of the complex. After 30 s of incubation, the grids were plunge-frozen in liquid ethane using a Vitrobot mark IV (FEI) with 95% chamber humidity at 10 °C.

**Electron microscopy**. Cryo-EM imaging was performed on a Titan Krios microscope operated at 300 kV (FEI) equipped with a Gatan K2 Summit direct electron camera (Gatan), a Gatan quantum energy filter, and a CS corrector (FEI). Images were recorded in EFTEM nanoprobe mode with Serial EM[48] in super-resolution counting mode with a super resolution pixel size of 0.55 Å and a defocus range of −1 to −3.2 μm. Six datasets were collected with a dose rate of 6–8 e⁻/pixel/s (1.1 Å pixel size at the specimen) on the detector. Images were recorded with a total dose of 50 electrons/Å², exposure time between 7–10 and 0.2–0.25 s subframes (35–50 total frames).

**Data processing**. Processing of each data set was done separately following the same procedure until the 3D refinements where particles were merged. The gain reference for the super-resolution dose-fractionated subframes was generated of each dataset with 'relion_estimate_gain' subprogram in RELION 3.1[49]. The subframes were then gain-corrected, binned twice, drift-corrected and dose-weighted using RELION 3.1 MotionCor2 own implementation yielding summed images with 1.1 Å pixel size. The contrast transfer function of the corrected micrographs was estimated using GCTF v1.06[50]. Thon rings were manually inspected for astigmatism and micrographs with measured resolutions worse than 5 Å were excluded. Particles were automatically picked by template matching in RELION2[51,52]. To generate the templates, around 6000 particles were manually picked on micrographs from the first data set using EMAN2[53]. Then, the manually picked particles were subjected to 2D classification in RELION2 and the best class averages were used as templates for subsequent automatic picking procedure. Taking together the six data sets, a total of 1,908,092 particles were selected from 13,484 micrographs. Particles from each data set were separately subjected to two rounds of 2D classification in RELION2 to remove junk particles and contaminations resulting in a total of 505,681 particles for further processing (Supplementary Fig. 1e). Particles from the six data sets were merged into two independent data sets that were each subjected to two rounds of 3D ab-initio classification in cryoSPARC v0[54] with a

class probability threshold of 0.9. After discarding the poor-quality models, the remaining particles were merged, resulting in a final data set of 162,332 particles. This final data set was used to generate a high-quality ab-initio model with cryoSPARC v0 (Supplementary Fig. 2). The ab-initio model was low-pass filtered to 30 Å and was used as a reference for 3D auto-refinement in RELION 3.1 producing a map of 7.0 Å global resolution. The refined particle coordinates were re-extracted and centered. This new particle stack was subjected to a 3D auto-refinement in RELION 3.1 using the ab-initio model low-pass filtered at 30 Å resulting in a 6.6 Å map (Supplementary Fig. 3). Local resolution calculated with Blocres[55] showed a range of resolution from 4 Å in the DNA-binding/cleavage domain and 14 Å in the ATPase domain indicating a high flexibility of the head. Moreover, 2D classifications showed large movements of the ATPase domain relative to the DNA-binding/cleavage domain (Supplementary Fig. 1e).

A combination of local approaches was used to identify different conformations and to improve local resolution of each domain. Since the ATPase head domain was too small for a 3D focused refinement, we first performed a focused 3D classification of the ATPase domain with a soft mask and no alignment in RELION 3.1. One class of 36,610 particles with well-defined densities of the ATPase domain was selected. To facilitate an accurate alignment of the particles, a new ab-initio model of the class was calculated in cryoSPARC. After aberration, magnification and per-particle CTF refinement in RELION 3.1 followed by two rounds of Bayesian polishing, homogeneous refinement in cryoSPARC v3[56] yielded a reconstruction with overall resolution of 7.6 Å in which density of the linkers appeared (Supplementary Fig. 3). The map was slightly better than the one refined in RELION 3.1.

Secondly, a focused 3D auto-refinement was performed in RELION 3.1 using a soft mask around the DNA-binding/cleavage domain yielding a 4.5 Å resolution reconstruction. Then, the particle stack was subjected to aberration, magnification and per-particle CTF refinement in RELION 3.1 followed by two rounds of Bayesian polishing. A focused 3D classification of the DNA-binding/cleavage domain starting with a fine angular sampling of 3.7°, local angular search range of 5° and tau fudge of 16 was performed. Two of the classes showed better angular accuracies and distinct conformations of the DNA-binding/cleavage domain, referred to as State 1 (60,601 particles) and State 2 (39,368 particles). After a last round of Bayesian polishing, these two classes were further refined in cryoSPARC v3 by homogeneous refinement with a C2 symmetry and gave a reconstruction with global resolution of 3.6 and 4.1 Å, respectively. Reconstructions with relative positions of the ATPase domain regarding both State 1 and State 2 conformations of the DNA-binding/cleavage domain were obtained by a first heterogeneous refinement followed by a homogeneous refinement in cryoSPARC v3 with a C1 symmetry. Auto-masking was disabled to avoid appearance of artifacts during refinement. The reconstruction of the overall complex with the DNA-binding/cleavage domain in State 1 conformation (26,506 particles) yielded a map of 4.6 Å overall resolution. For the DNA-binding/cleavage domain in State 2 conformation (13,420 particles), the reconstruction yielded a map of the entire complex at 7.6 Å overall resolution. Further heterogeneous refinements were performed to assess the flexibility of the ATPase domain with respect to the DNA-binding/cleavage domain (Supplementary Fig. 3).

All reported resolutions are based on the gold standard FSC-0.143 criterion[57] and FSC-curves were corrected for the convolution effects of a soft mask using high-resolution noise-substitution[58] in RELION 3.1 as well as in cryoSPARC v3. All reconstructions were sharpened by applying a negative B-factor that was estimated using automated procedures[59]. Local resolution of maps reconstructed in cryoSPARC v3 were calculated using Blocres (Supplementary Fig. 4). The cryo-EM maps of the DNA-binding/cleavage domain in State 1 and State 2 state as well as the entire complex in the two same conformations have been deposited in the EMDataBank under accession numbers EMD-11550, EMD-11551, EMD-11553, EMD-11554, respectively. The EM map with well-resolved N-gate/DNA-gate linkers have also been deposited in the EMDataBank under accession number EMD-11552.

**Model building and refinement of the DNA-binding/cleavage domain**. The two reconstructions of the DNA-binding/cleavage domain in State 1 and State 2 at 3.6 and 4.1 Å, respectively, were used to refine a crystal structure of the hTopo IIα DNA-binding/cleavage domain in complex with DNA and etoposide[27]. PDB 5GWK was stripped of all ions and water molecules, with all occupancies set to 1 and B-factors set to 50. First, the atomic model was rigid-body fitted in the filtered and sharpened maps using Chimera[60]. A first round of real-space refinement in PHENIX[61] was performed using local real-space fitting and global gradient-driven minimization refinement. Then, nucleic acids were modified according to the DNA sequence used in our structure. The visible part of the CTD linker (1187–1215) was built as a poly-A coil, as the quality of the EM density did not allow us to clearly attribute the register of the residues. Several rounds of real-space refinement in PHENIX using restraints for secondary structure, rotamers, Ramachandran, and non-crystallographic symmetry were performed, always followed by manual inspection in COOT[62], until a converging model was obtained. All refinement steps were done using the resolution limit of the reconstructions according to the gold standard FSC-0.143 criterion[57]. Refinement parameters, model statistics and validation scores are summarized in Supplementary Table 1. The atomic models of the DNA-binding/cleavage domain in State 1 and State 2 conformations have been

deposited in the Protein Data Bank under accession numbers 6ZY5, 6ZY6, respectively.

**Model building and refinement of the overall complex**. For both conformations of the DNA-binding/cleavage domain, the 3D reconstructions of the overall complex were used for further atomic model refinement. The atomic models previously refined for each conformation of the DNA-binding/cleavage domain were rigid-body fitted in the overall maps using Chimera. Then, crystal structure of the ATPase domain in complex with ADPNP was rigid-body fitted in the filtered and unsharpened maps using Chimera. PDB 1ZXM[8] was stripped of all ions and water molecules, with all occupancies set to 1 and *B*-factors set to 50. A first round of real-space refinement in PHENIX was performed using rigid-body and global gradient-driven minimization refinement. Then, the linker between ATPase domain and the DNA-binding/cleavage domain was built in COOT as an alpha helix, following the density and according to the secondary structure prediction (Supplementary Fig. 6). Refinement followed the same procedure as for the masked DNA-binding/cleavage domain except that the local real-space fitting was replaced by a rigid-body refinement. Resolution limit for refinements was set according to the gold standard FSC-0.143 criterion. Refinement parameters, model statistics, and validation scores are summarized in Supplementary Table 1. The atomic models of the full-length hTopo IIα in State 1 and State 2 conformations have been deposited in the Protein Data Bank under accession numbers 6ZY7, 6ZY8, respectively.

**Expression and purification of hTopo IIα mutants**. The modified pVote0GW vector used for wild-type hTopo IIα overexpression was mutated by site-directed mutagenesis using the QuikChange XL Site-Directed Mutagenesis kit (Agilent) in order to generate the plasmids harboring $K_{418}A$, $K_{425}A$, $W_{414}A$-$F_{417}S$, $K_{425}G$-$K_{426}G$-$C_{427}P$ mutations and Δ1193 or Δ1217 truncations. The primer used for the mutagenesis are available in the Supplementary information (Supplementary Table 3). The overexpression and purification procedure for the six mutants are identical to the wild type hTopo IIα described above in the "Methods" section, except for the truncated proteins Δ1193 or Δ1217 where the Strep buffer contained only 100 mM NaCl and 100 mM KCl.

**Relaxation assay**. An increasing concentration of hTopo IIα was incubated at 37 °C with 6 nmoles of supercoiled pUC19 plasmid in a reaction mixture containing 20 mM Tris–acetate pH 7.9, 100 mM potassium acetate, 10 mM magnesium acetate, 1 mM DTT, 1 mM ATP, 100 μg/ml BSA. After 30 min, reactions were stopped by addition of SDS 1%. Agarose gel electrophoresis was used to monitor the conversion of supercoiled pUC19 to the relaxed form. Samples were run on a 1 % agarose, 1× Tris borate EDTA buffer (TBE) gel, at 6 V/cm for 180 min at room temperature. Agarose gels were stained with 0.5 mg/ml ethidium bromide in 1× TBE for 15 min, followed by 5 min destaining in water. DNA topoisomers were revealed using a Synergy U:Genius 3 scanner.

**Cleavage assay**. An increasing concentration of hTopo IIα was incubated at 37 °C with 6 nmoles of supercoiled pUC19 plasmid in a reaction mixture containing 20 mM Tris–HCl pH 7.9, 50 mM potassium acetate, 10 mM magnesium acetate, 1 mM DTT, 1 mM ATP, 100 μg/ml BSA, and 275 μM of etoposide. Assays for the truncated proteins Δ1193 or Δ1217 and the WT hTopo IIα control were performed in presence and in absence of 1 mM ATP. After 30 min, reactions were stopped by addition of SDS 1% and 650 μM of proteinase K. After incubation at 37 °C 40 min, a phenol–chloroform DNA extraction at pH 8 was performed. Agarose gel electrophoresis was used to monitor the conversion of supercoiled pUC19 to the cleaved form. Samples were run on a 1% agarose, 1× TBE gel containing 0.5 mg/L of ethidium bromide (Euromedex), at 6 V/cm for 180 min at room temperature. Agarose gels were washed 5 min in water. DNA topoisomers were revealed using a Synergy U:Genius 3 scanner.

**ATPase enzymatic assays**. ATP hydrolysis was measured by following the oxidation of NADH mediated by pyruvate kinase (PK) and lactate dehydrogenase (LDH). The absorbance is monitored at 340 nm over 200 s at 37 °C with a Shimadzu 1700 spectrophotometer. Reactions were performed with 75 nM protein (240 nM for the triple mutant $K_{425}G$-$K_{426}G$-$C_{427}P$) and with or without 21 nM plasmid DNA (PUC19) in 500 μl of a buffer containing 50 mM Tris–HCl pH7.5, 150 mM potassium acetate, 8 mM magnesium acetate, 7 mM BME, 100 μg/mg of BSA, 4U/5U of PK/LDH mixture, 2 mM PEP, and 0.2 mM NADH and 4 mM ATP.

**Phylogenetic and structure prediction analysis of the TOP2**. 33 TOP2 genes, from unicellular eukaryotes to Human (Supplementary Table 4), were aligned with ClustalW[63] using the BLOSUM weight matrix. The subsequent alignment was injected into the ConSurf server[64] to analyze the conservation of the primary sequence. Secondary structure prediction of the hTopo IIα was performed using the PSIPRED server[65]. Protein domain graphs (Figs. 4 and 5) were generated using domainsGraph.py (archived at https://github.com/elifesciences-publications/domainsGraph)[66].

**hTopo IIα CTD production and purification**. The sequence coding for the human Topo IIα CTD (1191–1531) was inserted into a modified pAX vector containing an N-terminal 10 His-tag and a C-terminal twin-strep tag. Overexpression was performed in *E. coli* BL21 (DE3) pRARE2 in LB medium containing 50 μg/ml kanamycin and 35 μg/ml chloramphenicol. Cells were induced with 1 mM IPTG after reaching an $OD_{600}$ 0.95 and protein was expressed at 37 °C for 4 h. The CTD was purified similarly as described for the full-length human Topo IIα with few modifications: adjusted buffers (Lysis buffer: 25 mM Hepes, 500 mM NaCl, 500 mM KCl, 20 mM imidazole, 10% v/v glycerol, 1 mM PMSF, protease inhibitor cocktail (Roche), pH 7.5; Strep elution buffer: 25 mM Hepes, 200 mM NaCl, 200 mM KCl, 10% v/v glycerol, 2 mM DTT, 3 mM Desthiobiotin (Sigma), pH 7.5; Heparin elution buffer: 25 mM Hepes, 750 mM NaCl, 750 mM KCl, 10% v/v glycerol, 2 mM DTT, pH 7.5), an elution from the Heparin column by gradient (20 CV) and an additional size-exclusion chromatography step. After the ion exchange chromatography, fractions containing the intact CTD were pooled and loaded on a Superdex S200 16/60 size-exclusion chromatography column (GE Healthcare) using 25 mM Hepes, 100 mM NaCl, 100 mM KCl, 1 mM EDTA, 2.5% v/v glycerol, 5 mM DTT, pH 7.5. After the purification process, 4 mg of the hTopo IIα CTD were obtained per liter of cell culture (Supplementary Fig. 9a). Uncropped gel is provided in Source Data.

**SAXS experiments**. SAXS data of the hTopo IIα CTD (1191–1531) at 10 or 20 mg/ml was collected on an in-house Rigaku BioSAXS-1000 (Rigaku) apparatus, equipped with a Rigaku MicroMaxTM-007HF generator and a Pilatus 100k detector. The sample was maintained at 10 °C and exposed to the X-ray beam for 2 h. A total of eight images were recorded every 15 min. All the data processing steps were performed using the program package PRIMUS[67].

**Reporting summary**. Further information on research design is available in the Nature Research Reporting Summary linked to this article.

## Data availability

The data that support this study are available from the corresponding author upon reasonable request. All structural data and corresponding cryo-EM maps that support the findings of this study have been deposited to PDB and EMDB under the accession codes 6ZY5 and EMD-11550 for the hTopo IIα DNA-binding/cleavage domain in closed state; 6ZY6 and EMD-11551 for the hTopo IIα DNA-binding/cleavage domain in pre-open state; 6ZY7 and EMD-11553 for the entire hTopo IIα in closed state; 6ZY8 and EMD-11554 for the entire hTopo IIα in pre-open state; and EMD-11552 for the N-gate/DNA-gate linkers. Source data are provided with this paper.

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

## Acknowledgements

We thank Julio Ortiz and Gabor Papai for their help with EM data collection, as well as Marlène Vayssières for technical assistance with protein preparation. Computational resources were provided by the Méso-centre de Calcul (University of Strasbourg). We are also grateful to Albert Weixlbaumer and Sebastian Klinge for sharing computational resources. This work was supported by the Fondation ARC, the Association Alsace contre le Cancer and the grant ANR-10-LABX-0030-INRT (managed by the Agence Nationale de la Recherche under the frame program Investissements d'Avenir ANR-10-IDEX-0002-02). The authors acknowledge the support and the use of resources of the French Infrastructure for Integrated Structural Biology FRISBI ANR-10-INBS-05 and of Instruct-ERIC.

## Author contributions

A.V.B. and V.L. conceived the study and designed the experiments; A.V.B., C.L., R.D., L.H., and C.B. performed the experiments; A.V.B. performed the cryo-EM data collection, processed the cryo-EM data and built the atomic models; A.V.B., C.L., and V.L. analyzed and interpreted the data; A.V.B., C.L., and V.L. wrote the manuscript.

## Competing interests

The authors declare no competing interests.
