## [Peer Review File · Nature Communications]

Reviewers' Comments:

Reviewer #1:

Remarks to the Author:

The manuscript by Lamour et al. describes the cryo-EM structure of the complete human topoisomerase II alpha complex with DNA. It is the first cryo-EM structure of the complete enzyme and only the second structure of a kinetically competent eukaryotic type II enzyme (the first was the crystal structure of yeast topoisomerase II that lacked only its C-terminal domain). The work is timely and addresses important issues in the field. The authors provide structural evidence for a novel regulatory element in human topoisomerase II alpha that links the DNA binding/cleavage domain and the C-terminal domain. Overall, the work by the Lamour laboratory is nicely done, and makes an important contribution to the topoisomerase field. The manuscript should definitely be of interest to readers of Nature Communications. However, a few issues need to be addressed prior to acceptance. Specific comments follow

1. The ATPase assays performed in the manuscript were carried out in the presence of DNA. However, topoisomerase II will also hydrolyze ATP in the absence of nucleic acids. It would be interesting to repeat the ATPase experiments in the absence of ATP to see if the effects that are described represent a change in the ATPase domain or an allosteric effect that depended on the presence of DNA. It would strengthen some of the key arguments in the manuscript.
2. The authors state that, as observed in previous studies, the DNA relaxation activity of human topoisomerase II alpha that lacked its C-terminal domain (delta 1193) was similar to that of the wild-type enzymes. However, the cited reference (ref 36) indicates that the catalytic activity of human topoisomerase II alpha lacking its C-terminal domains is reduced by ~2-fold.
3. The authors indicate that the DNA cleavage response to etoposide by their mutant topoisomerase II alpha that lacked a C-terminal domain (delta 1193) was diminished and stated that this was similar to data shown in ref 36 (which is correct). However, that earlier work also demonstrated that in the absence of drug, levels of DNA cleavage of the human enzyme that lacked its C-terminal domain was 2-3 times higher than that of wild-type topoisomerase II alpha. The authors also found a large increase in baseline DNA cleavage by their mutant enzyme. Do the new structural findings by the authors shed any light on this increase in DNA cleavage.
4. Because the authors include ATP in their DNA cleavage assays, it is not intrinsically possible to distinguish the linear DNA band, which represents the cleavage product, from relaxed DNA topoisomers. The data shown would be much cleaner if the authors included ethidium bromide in their gels (which will collapse all of the covalently closed DNA topoisomerase into the supercoiled band), to be able to more easily (and demonstratively) identify and quantify the DNA cleavage product. It also might be worthwhile for the authors to examine DNA cleavage in the absence of ATP to determine whether the effects of the C-terminal domain on scission take place prior to DNA strand passage.
5. The authors indicate that partial deletions of the C-terminal domain have not been examined previously. This is not the case; ref 34 examined several partial deletions of the C-terminal domain, including delta 1321. The previous work did not find an increase in DNA relaxation activity associated with this deletion mutant, whereas the current work indicates a ~10-fold increase in relaxation rates with delta 1217 compared to wild-type. Can the authors provide any reasons for the difference in findings?

Reviewer #2:

Remarks to the Author:

Vanden Broeck and colleagues present the structure of human topoisomerase 2alpha (hTop2a) by

cryoEM together with biochemical information to validate some of their structural findings. The structure of the intact hTop2a was not known before, but the structures of the ATPase domain and the cleavage/religation domains were known as well as the structure of the intact gyrase, a closely related bacterial topoisomerase 2. The authors set out to solve the structure of the intact protein, including the C-terminal domain (CTD), in complex with a short DNA fragment. Their findings are almost in complete agreement with what was known before. There are few surprises here. The two domains are arranged in the intact enzyme in the expected way and with the overall arrangement as previously observed in gyrase. The C-terminal domain is invisible in the structure and only a short linker is ordered. The structure confirms many of the previous observations and conclusions and adds limited new information. The biochemical work supports the structural observations, but again there are no major surprises or findings. Overall, this is a very nice piece of work, but most of the findings are not unexpected or unanticipated and provide limited new information on the mechanism or biological function of the enzyme. There are some specific concerns listed below.

1. The cryoEM reconstructions are to medium resolution. Calling them "near-atomic" is misleading.
2. The authors conclude that the CTD is not folded based on its absence in the cryoEM reconstructions, sequence-based prediction, and SAXS data on a CTD fragment expressed in bacteria. I am not convinced that the SAXS data are relevant without more controls. The CTD used could have been unfolded due to its expression in bacteria. There is simply not enough information to conclude conclusively that the CTD is disordered. It is likely to be, but more information is needed.
3. The biochemical experiments support well the observations, but showing gels is not enough. Some quantitation based on the gels is needed. It is difficult to assess the level of changes in supercoiling activity based on gels.

Reviewer #3:

Remarks to the Author:

The manuscript by Vanden Broeck described the cryo-EM structures of the full-length human Topo2a bound to the anticancer drug etoposide. The authors determined the structure of the DNA binding/cleavage domain in two different conformations (closed and pre-open) at 4.2 Å and 5.7 Å, which allowed the previous crystal structure to be fitted and refined into the map. The ATPase domain is highly flexible in the structure. Taking advantage of focused refinement/classification methods in cryoEM, they analyzed the dynamics and obtained a reconstruction containing both the ATPase and DNA binding domain at 7.6 Å resolution. Based on the secondary structure prediction, the linker between the two domains was modelled as a long helix. Comparison of the pre-opening and closed states revealed movements in both domains. The role of the linker helix between the two domains was further investigated. Through sequence alignments, the authors identified two highly conserved motifs (W414xxF417K418 and K425K426C427) within the linker and studied their effects on the DNA relaxation, cleavage and ATPase activities of the enzyme. Finally, in the 4.2Å map, an additional density was assigned to the beginning of the disordered C-terminal domain. Using different C-terminal truncations, it was found that this assigned fragment of the CTD stimulated the relaxation and ATPase activities of the enzyme.

In general, while I found the work presented in this manuscript is solid and interesting, the resolutions of the EM maps here are still limited for some of conclusions described, and there seems to be a lack of discussions relating the findings described here to the mechanism of how Topo2a acts to relax super-coil DNA. Below are a number of specific points that should be addressed.

Major points

1. So far all the focused 3D classification/refinement has been focused on the DNA binding domain, it was stated that the ATPase domain is too small for this. RELION2.0 is used for processing. RELION has since been improved significantly with the latest 3.0 and 3.1 versions with many features including multibody refinement, signal subtraction with recentering/reboxed, Bayesian polishing, aberration corrections etc. All these will likely allow the author to resolve the ATPase domain and its conformation

dynamics better and improve resolution of all reconstructions shown in the structure including the DNA binding domain, ATPase domain and overall structure significantly. The better resolution will allow confidence in assignment of the linker region and the CTD as well.

2. The authors proposed a coupling mechanism through the 27aa linker between the two domains. This linker is modelled as an alpha helix based on the SS structure prediction. At 7.6 Å resolution, the residue assignment for this can be tricky. Additional validation by other means for this assignment would be needed, especially if the authors were to show side-chains in Fig. 4d and suggest mechanistic details relating motifs in this helix. Also by using the newer RELION version, the resolution for this may be improved to an extent that large aromatic side chains are visible.

3. The choices of the consensus sequence motifs of this linker need more experiments and clarification. From Fig. 4b, the defined levels of conservation mentioned in the text seem inconsistent. Except for K425, which is conserved, the others are not all conserved at all in the lower eukaryotes. E.g. the WxxFK motif. At the same time, it was mentioned that K418 is strictly conserved and it was close to the K-loop in the yeast protein. This residue and motif are not conserved in yeast, would one expect it to have a similar role between human and yeast? W and F are very large side-chains and in lower eukaryotes, they are I/Q/T or A/G, which are much smaller. This is also confusing together with the activity assays, where W414A-F427S and K418A show little change compared to the WT. This is not surprising because they are not really conserved. The major difference in the activity between the WT and the mutants are in the K425K426C427 mutant. This makes sense because K425 is completely conserved. Like the WxxFK motif, K426 and C427 are not really conserved. Could this effect be caused by the single K425A mutation alone? This single mutation should be tested to dissect the contribution of the non-conserved residues.

4. Furthermore, there is another motif at the beginning of the linker, GIVE (406-409), which has a similar level of conservation to the WxxFK. However, it was not included in the mutagenesis and commented on why it was omitted.

5. Part of the CTD domain was assigned (1192-1215). Based on Supplemental Fig. 5g, there is a density break between the coiled coil region and the assigned stretch. This can make the residue assignment uncertain if there is a disordered loop of unknown length in between. Can the authors provide additional data/validation on this residue assignment?

6. Based on deletion mutants, this region is proposed to be a positive regulator of the relaxation activity. In the context of the FL construct, there is an autoinhibitory mechanism involved? If so, how would this be released to allow for the positive influence of this part of the CTD?

Minor points

7. The term "near-atomic resolution" is debatable these days. Many have argued that atomic would be 1Å, near-atomic could actually be 1-2 Å rather than 4-5 Å we see in EM (See Wlodawer & Dauter Acta Cryst D. 2017 for more details) (Line 89)

8. The EM density fit shown in Fig. 1d and Supp. Fig. 5f for etoposide is not very clear as stated in the manuscript. Any comments on this? Perhaps a better view or better of showing this will be more convincing.

9. Line 111-122, the authors discussed the cavity to accommodate T-segment on top of G-segment, C-gate, N-gate etc and citing Fig. 3. Fig. 3 could be improved to label these features clearly for clarity. For an audience not familiar the topoisomerase field, it is very difficult to follow these terms without an illustration. This is applied for the remainders of the figures and texts.

10. Fig. 4e, F427S is a mistake?

11. Line 221, prokaryotic

Title: Structural basis for allosteric regulation of Human Topoisomerase 2 α

Authors: Vanden Broeck Arnaud, Lotz Christophe, Robert Drillien, Léa Haas[£], Claire Bedez, Lamour Valerie

[£] Author added during the revision process.

Point by point answer to reviewers' comments

We thank all the reviewers for the time they have taken to carefully review our work and for their comments, which helped us to improve our manuscript. We have addressed each reviewer comment (shown in black bold) in our point-by-point response below, where we respond in blue text, with changes highlighted in grey the main text document and the supplementary information.

Reviewers' comments:

Reviewer #1 (Remarks to the Author):

The manuscript by *Lamour et al.* describes the cryo-EM structure of the complete human topoisomerase II alpha complex with DNA. It is the first cryo-EM structure of the complete enzyme and only the second structure of a kinetically competent eukaryotic type II enzyme (the first was the crystal structure of yeast topoisomerase II that lacked only its C-terminal domain). The work is timely and addresses important issues in the field. The authors provide structural evidence for a novel regulatory element in human topoisomerase II alpha that links the DNA binding/cleavage domain and the C-terminal domain. Overall, the work by the Lamour laboratory is nicely done, and makes an important contribution to the topoisomerase field. The manuscript should definitely be of interest to readers of Nature Communications. However, a few issues need to be addressed prior to acceptance. Specific comments follow

We thank the reviewer for the appreciation of our work and for the time taken to review our manuscript. We have answered below the remarks and comments.

1. The ATPase assays performed in the manuscript were carried out in the presence of DNA. However, topoisomerase II will also hydrolyze ATP in the absence of nucleic acids. It would be interesting to repeat the ATPase experiments in the absence of ATP to see if the effects that are described represent a change in the ATPase domain or an allosteric effect that depended on the presence of DNA. It would strengthen some of the key arguments in the manuscript.

We thank the reviewer for this useful suggestion. We have repeated the ATPase assays in presence and absence of DNA for the 7 proteins on freshly purified samples (WT + 5 original mutants + 1 new mutant suggested by reviewer #3). Multiple replicates on different protein preparations, and not just technical replicates, allowed us to normalize the assays among the different mutants and WT. As a consequence, some values may differ slightly from the original version but still showing the same trend relative to each other within error margins.

As one could expect from previously published work, all protein samples displayed a reduced ATPase activity in absence of DNA as it was shown to stimulate ATP hydrolysis. However, the triple mutant in the transducer helices, K425G-K426G-C427P, displayed a stronger decrease in ATPase activity in absence of DNA showing that this area is critical for allosteric signaling (Figure 4c and Supplementary Fig. 9e). We have modified the result section accordingly.

2. The authors state that, as observed in previous studies, the DNA relaxation activity of human topoisomerase II alpha that lacked its C-terminal domain (delta 1193) was similar to that of the wild-type enzymes. However, the cited reference (ref 36) indicates that the catalytic activity of human topoisomerase II alpha lacking its C-terminal domains is reduced by ~2-fold.

We thank the reviewer for their remark. We found that there was an error in the reference we cited at this location in the text. We intended to compare our results to the work of Kawano et al. on the truncated rat TopoII α comprising residue 1 to 1191 (doi: 10.1093/jb/mvv110). They tested the relaxation activity on a negative supercoil and also observed that the activity of their mutant is similar to the WT. We repeated the experiments as part of this revision and after additional replicates, we confirm that relaxation activity of the human 1193 CTD-truncation is similar to the WT (but the cleavage activity with etoposide is reduced 5-fold and ATPase activity is reduced about 30%).

In addition, the article by Dickey et al. that was previously cited (reference 36, doi:10.1021/bi050811i) did not analyze the truncation at position 1193 but a larger truncation from residue 1175. Truncation in this area shortens an α -helix that folds against the core of the DNA-binding cleavage domain, while truncation at 1193 preserves the entire α -helix. The 1175 truncation may impair the structural stability of the DNA-binding cleavage domain. We see in our analysis that enzymatic activities of the Top2 are quite sensitive to the alteration of secondary structures elements part of, or nearby, the DNA binding site.

3. The authors indicate that the DNA cleavage response to etoposide by their mutant topoisomerase II alpha that lacked a C-terminal domain (delta 1193) was diminished and stated that this was similar to data shown in ref 36 (which is correct). However, that earlier work also demonstrated that in the absence of drug, levels of DNA cleavage of the human enzyme that lacked its C-terminal domain was 2-3 times higher than that of wild-type topoisomerase II alpha. The authors also found a large increase in baseline DNA cleavage by their mutant enzyme. Do the new structural findings by the authors shed any light on this increase in DNA cleavage.

We performed again all the assays for the revisions and we proceeded to quantify all the bands. We now have a more accurate assessment of the activities of the mutants relative to the WT. We could observe that the mutant enzyme Δ 1193 is indeed less active than the WT in cleavage assays with etoposide (Figure 5 and Supplementary Fig. 9). Our structure suggests that the 1193-1217 linker could stabilize the etoposide binding site, and that its absence in the 1193 deletion is detrimental to the formation of the etoposide cleavage complex.

We are not sure to which panel the reviewer is referring to when he mentions “a large increase in baseline DNA cleavage by their mutant enzyme”. It is more difficult to assess the intensity of the baseline cleavage in our relaxation assay without proper control, as it was not initially designed to analyze cleavage. If we however consider that the Δ 1193 mutant shows a larger increase in baseline DNA cleavage than the WT, then it means the CTD may exert a different regulation in presence or absence of etoposide. It could be that the presence of etoposide is restricting or allowing specific conformational arrangement of the CTD relative to the catalytic core, therefore limiting cleavage.

To answer this question, it would be necessary to identify the location of the bulk of the CTD. The present structures already allowed us to identify the linker region leading to the CTD and to analyze the impact of this region on the catalytic activities. The position of the linker indicates that the CTD would be protruding in an orthogonal orientation relative to the axis of the G-segment. Based on this first assessment, one could imagine designing FRET experiments with adequate positioning of acceptor-donor molecules to probe the CTD position relative to catalytic core during the catalytic cycle. Ultimately, we hope that the structure of the complete CTD may be obtained in specific conditions, which still remain to be established.

4. Because the authors include ATP in their DNA cleavage assays, it is not intrinsically possible to distinguish the linear DNA band, which represents the cleavage product, from relaxed DNA topoisomers. The data shown would be much cleaner if the authors included ethidium bromide in their gels (which will collapse all of the covalently closed DNA topoisomerase into the supercoiled band), to be able to more easily (and demonstratively) identify and quantify the DNA cleavage product. It also might be worthwhile for the authors to examine DNA cleavage in the absence of ATP to determine whether the effects of the C-terminal domain on scission take place prior to DNA strand passage.

We are grateful for the technical suggestion regarding the electrophoretic conditions of cleavage assays. We agree that distinguishing the cleavage band was previously difficult and could have been misleading in particular since the impact of some mutations can be subtle. We added ethidium bromide in our gel and performed the experiments in triplicates for all 7 protein samples to be able to quantify properly the cleavage band relative to the remaining negatively supercoiled DNA (Figure 4f & 5f, Supplementary Fig. 9 and Source Data). This has greatly improved both the reproducibility of the assay and the interpretation of the data. As also suggested by reviewer 2, we quantified the relaxation assays and provided graphical figures instead of gel images. The analysis of the assays has been completely rewritten in the Results section (pages 7-10) and we have modified Figure 4 and 5 accordingly. Raw images for the cleavage and relaxation assays have been provided as Source Data file.

As suggested, we also performed the cleavage assay in absence of ATP for the C-terminal mutants and the control WT sample (Supplementary Fig. 9d). This results in the interesting observation that even in absence of ATP (hence in absence of N-gate dimerization), the 1217 truncation keeps about 80% of its cleavage activity in presence of etoposide when the WT and 1193 truncation show a sharp decrease to about one fifth of the initial level. Our structural data shows that the region 1200 to 1217 stretches underneath the DNA helix, and may contribute to maintain the DNA curvature, favoring DNA cleavage during DNA relaxation and upon etoposide binding. The similar activities of the WT and of the 1193 truncation means that the stimulation of the DNA relaxation activities and of DNA cleavage can be attributed to the linker, but is counterbalanced by the bulk of the CTD in the full length protein.

As the reviewer suggests, this could imply that effects of the CTD on scission could potentially take place prior to DNA strand passage. It remains however difficult at this stage to describe how the effect of the CTD could happen at the structural level without further information on the location of the bulk of the CTD, relative to the DNA-binding/cleavage site.

5. The authors indicate that partial deletions of the C-terminal domain have not been examined previously. This is not the case; ref 34 examined several partial deletions of the C-terminal domain, including delta 1321. The previous work did not find an increase in DNA relaxation activity associated with this deletion mutant, whereas the current work indicates a ~10-fold increase in relaxation rates with delta 1217 compared to wild-type. Can the authors provide any reasons for the difference in findings?

We indeed cited the prior studies of reference 34 examining other truncations (line 223, doi: 10.1021/bi800453h). We were not implying in the text that partial deletions were never examined previously. We mentioned that the linker (region 1193-1217) was never observed in a molecular structure, since all available structures are always truncated at position 1193:

- page 9 line 221: "the linker leading to the CTD has not been previously observed in a eukaryotic Top2 structure".
- page 9 line 226: "However, most studies used a construct of Top2 ending before, or at position 1192, therefore not accounting for the contribution of this linker region that is in close proximity with the G-segment³⁷."

To properly cite this reference, we modified the text by adding in page 9 line 226: "Although previous studies have examined the effect of CTD deletion at different positions, the Top2 constructs were ending before, or at position 1192, therefore not accounting for the contribution of this linker region that is in close proximity with the G-segment³⁷."

McClendon *et al.* monitored the relaxation activity of truncation $\Delta 1521$, $\Delta 1501$, $\Delta 1476$, $\Delta 1321$, $\Delta 1175$ on negatively supercoiled DNA and concluded that the activities are similar to the WT. Truncations $\Delta 1321$ and $\Delta 1175$ behave like the WT regarding DNA relaxation which is consistent with our results on $\Delta 1193$ and the WT. However, the region spanning from residues 1193 to 1217 was not examined in their study. The truncation $\Delta 1175$ is too short and $\Delta 1321$ too long to distinguish the impact of this particular linker region. It's only because we have determined the molecular structure using the whole sequence of the Top2 that we were able to see the region between 1193 and 1217 and examine its effect. It would have been otherwise very difficult to predict that this region would behave as such and to design this truncation *ab initio*.

We show in our study that the 1193-1217 linker region of the CTD stimulates the overall activity of TopII α suggesting that the rest of the C-terminal domain down-regulates this effect.

Reviewer #2 (Remarks to the Author):

Vanden Broeck and colleagues present the structure of human topoisomerase 2alpha (hTop2a) by cryoEM together with biochemical information to validate some of their structural findings. The structure of the intact hTop2a was not known before, but the structures of the ATPase domain and the cleavage/religation domains were known as well as the structure of the intact gyrase, a closely related bacterial topoisomerase 2. The authors set out to solve the structure of the intact protein, including the C-terminal domain (CTD), in complex with a short DNA fragment. Their findings are almost in complete agreement with what was known before. There are few surprises here. The two domains are arranged in the intact enzyme in the expected way and with the overall arrangement as previously observed in gyrase. The C-terminal domain is invisible in the structure and only a short linker is ordered. The structure confirms many of the previous observations and conclusions and adds limited new information. The biochemical work supports the structural observations, but again there are no major surprises or findings. Overall, this is a very nice piece of work, but most of the findings are not unexpected or unanticipated and provide limited new information on the mechanism or biological function of the enzyme. There are some specific concerns listed below.

We thank the reviewer for taking the time to review our manuscript. We have responded below to all the remarks and comments.

To the best of our knowledge, the structure of the full-length human TopII alpha/beta has never been solved either by X-ray crystallography or by cryo-electron microscopy. The catalytic domains have been solved individually using X-ray crystallography, but they do not provide any information on the connections between the catalytic domains. The structure of the catalytic core of the yeast enzyme has been solved at 4.4 Å (doi: 10.1038/nsmb.2388). The connections between the ATPase domain and the DNA binding-cleavage domain were unfortunately not visible in the electron density, and the protein was truncated of its CTD extension (and which is not conserved in the human enzyme). The fact that we used the full-length construct to determine the architecture of the human TopII allowed us to discover and examine these connections that are crucial for its allosteric properties. In addition, we solved 2 distinct conformations of the full-length complex showing for the first time the relative movements of the catalytic domains.

We believe this is new information to the field as it finally provides a structural basis for analysis the allosteric mechanism. In addition, our molecular model now provides structural rationale to investigate human post-translational modifications that could regulate catalytic activities and drug resistance.

1. The cryoEM reconstructions are to medium resolution. Calling them “near-atomic” is misleading.

We agree that the cryo-EM reconstruction of the full-length Top2 is of medium resolution (4.7 Å) mostly due to the intrinsic flexibility of this molecular assembly. However, thanks to reviewer #3's suggestions for data processing, we were able to significantly improve the resolution on the most ordered regions. Notably, this allows us to distinguish the etoposide molecule at to a level of details that is now close to the standard of electron densities in X-ray crystallography (3 Å local resolution). We agree that "near-atomic resolution" might be an overstatement, but it was commonly used in the EM field to say "similar to what could be obtained using X-ray crystallography". We are aware that the EM field has been evolving extremely fast in the recent years and what was considered “near-atomic resolution” three years ago (3-4 Å) is now medium resolution. We modified this statement in the text whenever applicable.

2. The authors conclude that the CTD is not folded based on its absence in the cryoEM reconstructions, sequence-based prediction, and SAXS data on a CTD fragment expressed in bacteria. I am not convinced that the SAXS data are relevant without more controls. The CTD used could have been unfolded due to its expression in bacteria. There is simply not enough information to conclude conclusively that the CTD is disordered. It is likely to be, but more information is needed.

As a first assessment, cryo-EM reconstructions, sequence-based prediction, and SAXS data on the CTD fragment are good indications that this domain is disordered. It is worth noting that the cryo-EM reconstructions are based on protein produced in mammalian cells. Contrary to the catalytic core, the CTD cannot be seen in the 3D reconstruction, nor visualized in the 2D classes. The averaging process at these steps of the structural analysis could indeed explain why we might not see this domain if it was to adopt a mix of different conformations. However, we are not able to distinguish these 38 kD region even on raw images without any averaging, which would be indicative of a stretched and poorly structured domain. By comparison, the 47 kDa ATPase domain that is flexible is clearly visible at the level of raw images, 2D classes and 3D reconstruction.

We however agree that still more experiments could be performed, and we have modulated this conclusion in the result section (line 209-210): “Although complementary experiments would be required to conclude on the CTD structure, it suggests that in these conditions this domain is disordered or highly flexible.”

We believe that this domain may only adopt a definite structure when bound to other protein partners or chromatin and may also depend on the post-translational status during the cell cycle.

3. The biochemical experiments support well the observations, but showing gels is not enough. Some quantitation based on the gels is needed. It is difficult to assess the level of changes in supercoiling activity based on gels.

We thank the reviewer for this suggestion. We agree it is much easier to analyze the data with quantification and proceeded to the quantification of assays for all protein samples. As suggested by reviewer #1, we also quantified the cleavage assays and provided graphical figures instead of gel images for both assays. The analysis of the assays has been completely rewritten in the Results section (page 7-10) and we have modified Figure 4 and 5 accordingly. Raw images for the cleavage and relaxation assays have been provided as Source Data file.

Reviewer #3 (Remarks to the Author):

The manuscript by Vanden Broeck described the cryo-EM structures of the full-length human Topo2a bound to the anticancer drug etoposide. The authors determined the structure of the DNA binding/cleavage domain in two different conformations (closed and pre-open) at 4.2 Å and 5.7 Å, which allowed the previous crystal structure to be fitted and refined into the map. The ATPase domain is highly flexible in the structure. Taking advantage of focused refinement/classification methods in cryoEM, they analyzed the dynamics and obtained a reconstruction containing both the ATPase and DNA binding domain at 7.6 Å resolution. Based on the secondary structure prediction, the linker between the two domains was modelled as a long helix. Comparison of the pre-opening and closed states revealed movements in both domains. The role of the linker helix between the two domains was further investigated. Through sequence alignments, the authors identified two highly conserved motifs (W414xxF417K418 and K425K426C427) within the linker and studied their effects on the DNA relaxation, cleavage and ATPase activities of the enzyme. Finally, in the 4.2Å map, an additional density was assigned to the beginning of the disordered C-terminal domain. Using different C-terminal truncations, it was found that this assigned fragment of the CTD stimulated the relaxation and ATPase activities of the enzyme.

In general, while I found the work presented in this manuscript is solid and interesting, the resolutions of the EM maps here are still limited for some of conclusions described, and there seems to be a lack of discussions relating the findings described here to the mechanism of how Topo2a acts to relax super-coil DNA. Below are a number of specific points that should be addressed.

We express our gratitude to the reviewer for carefully reading our manuscript and providing a detailed review. We have taken into account all the remarks and suggestions which have considerably improved our manuscript.

Major points:

1. So far all the focused 3D classification/refinement has been focused on the DNA binding domain, it was stated that the ATPase domain is too small for this. RELION2.0 is used for processing. RELION has since been improved significantly with the latest 3.0 and 3.1 versions with many features including multibody refinement, signal subtraction with recentering/reboxed, Bayesian polishing, aberration corrections etc. All these will likely allow the author to resolve the ATPase domain and its conformation dynamics better and improve resolution of all reconstructions shown in the structure including the DNA binding domain, ATPase domain and overall structure significantly. The better resolution will allow confidence in assignment of the linker region and the CTD as well.

We thank the reviewer for giving us the opportunity to reprocess all our cryo-EM data. We took advantage of the new features in Relion 3.1 and cryoSPARC v3 to further improve the resolutions of our reconstructions. Using Bayesian polishing and CTF refinement, resolution of the DNA-binding/cleavage domain in closed and pre-open states has been improved to 3.6 Å and 4.1 Å (previously 4.1 Å and 5.7 Å), respectively. Here are details on the different parts of the structure we wanted to focus on and improve:

- The EM density for the etoposide, intercalating the DNA bases in the close vicinity of the cleavage site, has significantly improved. It is now visibly separated from the DNA bases density (as illustrated in the new Figure 1 and Supplementary Fig. 5). Our study now further illustrates the potential for drug discovery on Human topoisomerase II using cryo-EM.
- The region of the CTD linker has not improved sufficiently to allow us to distinguish the side chains and thus to assign the exact register. However, we believe this stretch can be attributed to residues 1193 to 1215. This point will be discussed further in reviewer's point 5.

- Regarding the ATPase domain, we were unfortunately unable to solve the structure at higher resolution. The highly flexible nature of this 90 kDa domain has made the alignment unsuccessful, at least in our hands (see the flexibility in Fig. 1, Fig 2 and Supplementary Fig. 3). We have tried to perform multi-body refinement: only the DNA-binding/cleavage domain could be aligned and refined. An attempt of signal subtraction with and without centering and reboxing on the ATPase domain was not successful either. However, thanks to the Bayesian polishing and CTF refinement, we were able to solve the overall structure of the Human topoisomerase II alpha in closed state at higher resolution (4.7 Å instead of 5.6 Å). A more stringent classification allowed us to find a particle stack with a better density of the ATPase domain, as illustrated in Fig. 2 and Supplementary Fig. 3. As the linker between the two linkers is also subjected to a lot of motions to accommodate the ATPase domain movements, we were unfortunately unable to improve this region either. This point is further discussed in reviewer's point 2.
- Thanks to the higher resolution on the closed conformation map (3.6 Å), we found Mg²⁺ density at the metal ion B site, complexed by the DxD motif. The absence of Mg²⁺ at site A confirms the etoposide-bound doubly-nicked DNA configuration present in our data.

2. The authors proposed a coupling mechanism through the 27aa linker between the two domains. This linker is modelled as an alpha helix based on the SS structure prediction. At 7.6 Å resolution, the residue assignment for this can be tricky. Additional validation by other means for this assignment would be needed, especially if the authors were to shown side-chains in Fig. 4d and suggest mechanistic details relating motifs in this helix. Also by using the newer RELION version, the resolution for this may be improved to an extent that large aromatic side chains are visible.

We agree with the reviewer that the density in this region of the EM maps do not allow the clear identification of side chains in the linker. Our rationale for building it as an alpha helix is based on the following elements:

- The beginning of the linker (residues 406-411) is folded as an alpha helix in the X-ray structure of the ATPase domain (PDB ID 1ZXM) (in red, figure below). This gives an anchor point and a direction to build the rest of the linker as an alpha helix. When doing so, a hydrophobic patch (WxxF) is formed at the interface of the helices and these residues are not exposed to the surface.
- The secondary structure prediction gives a high confidence score for alpha helix folding.
- The distance between the last residue in the X-ray structure of the ATPase domain (PDB ID 1ZXM) and the first residue of the DNA-binding/cleavage domain is such that all the residues of the linker need to fold into an alpha helix in order to accommodate this short distance.
- The EM density at 7.2 Å shows a "tube" shape which is compatible with the fold of an alpha helix. A random coil would not be observable at this resolution.
- Bacterial and eukaryotic Topo II share a conserved ternary/quaternary overall structure. The transducer linker is an alpha helix in bacteria.

Even though the resolution for this region is not high enough to distinguish side chains, we mapped the residues on the linker to provide the reader an approximative position and direction for these residues (Figure 4d). These new structural elements, together with the structure-guided mutagenesis and functional assays, allowed us to probe the potential role of the conserved residues in the context of the allosteric signal transduction in Topo II.

3. The choices of the consensus sequence motifs of this linker need more experiments and clarification. From Fig. 4b, the defined levels of conservation mentioned in the text seem inconsistent. Except for K425, which is conserved, the others are not all conserved at all in the lower eukaryotes. E.g. the WxxFK motif. At the same time, it was mentioned that K418 is strictly conserved and it was close to the K-loop in the yeast protein. This residue and motif are not conserved in yeast, would one expect it to have a similar role between human and yeast? W and F are very large side-chains and in lower eukaryotes, they are I/Q/T or A/G, which are much smaller. This is also confusing together with the activity assays, where W414A-F427S and K418A show little change compared to the WT. This is not surprising because they are not really conserved. The major difference in the activity between the WT and the mutants are in the K425K426C427 mutant. This makes sense because K425 is completely conserved. Like the WxxFK motif, K426 and C427 are not really conserved. Could this effect be caused by the single K425A mutation alone? This single mutation should be tested to dissect the contribution of the non-conserved residues.

We thank the reviewer for pointing out this lack of clarity regarding the sequence alignment and the identification of conserved residues. Our reasoning was the following:

1. To obtain a robust and broad sequence alignment, we first decided to align the TOP2 sequences from 35 different eukaryotic species (Supplementary Table 4). We picked a wide range of model organisms, including unicellular parasites, plants, yeasts, worms, insects, mollusks and many vertebrates.
2. After careful analysis of the alignment, we observed that some of the residues in the linker were extremely well conserved only among metazoans and multicellular plant species (31 out of the 35). All unicellular eukaryotes were actually quite divergent. This could be explained by the fact that these organisms are primitive and have evolved quite rapidly into the different clades of superior eukaryotes.
3. We then decided to only take into account the conservation pattern from the metazoans and plants, considering the unicellular eukaryotes being too divergent and having hybrid conserved motifs with Archaea and Bacteria. Moreover, it has been shown that topoisomerases in unicellular eukaryotic pathogens possess significant structural and functional differences from the multicellular ones (doi: 10.1016/j.tibs.2018.12.001).
4. The confusion came from the fact that the sequences alignment in Figure 4 still included unicellular species although our rationale for designing the mutations excluded them.

We have now edited Fig. 4 to reflect the conserved pattern only among multicellular plants and metazoans. We left the unicellular *S. cerevisiae* below the alignment as a comparison, as structures of yeast Top2 have been published, but unicellular species were not used in the new sequence alignment nor for the conservation calculation. We also added the complete alignment in the Supplementary Fig. 11 for the appreciation of the reader. When considering the plants and metazoans, which covers 800 million years of evolution (doi: 10.1098/rstb.2015.0036), it now highlights the high degree of conservation for residues W414, F417, K418 and K425 among species in this group. We also have revised the main text to reflect these changes and we have updated our explanations in the result section (page 6-7) and reorganized the discussion (page 10).

As requested by the reviewer, we also performed the mutagenesis of K425 into alanine. Even though this residue is conserved in superior eukaryotes, we did not observe any change in relaxation, cleavage or ATPase activities for the K425A mutant compared to the WT. Based on our structural data, it was not surprising to us as this residue is not involved in any interaction with other residues of the structure and is isolated on the transducer linker. It is still unclear why this residue is well conserved, but it does not seem to be involved in the transduction of the signal. The mutation of K425 alone has no detectable effect, but when the alpha helix is broken by the replacement of K425/K426/C427 by GGP, we observe a significant impairment in DNA relaxation and cleavage activities, highlighting the importance of the structural rigidity of the transducer helix linker. Similarly, the removal of the hydrophobic patch has a negative effect on the cleavage activity of the enzyme (in presence of etoposide), showing that the coordination and interaction of the two transducer linkers when the ATPase domain dimerize is crucial for the cleavage activity of the enzyme.

4. Furthermore, there is another motif at the beginning of the linker, GIVE (406-409), which has a similar level of conservation to the WxxFK. However, it was not included in the mutagenesis and commented on why it was omitted.

We thank the reviewer for raising this point. We did not consider this motif because it is known from the X-ray structure of the ATPase domain alone (PDB ID 1ZXM) that both I407 and V408 form a hydrophobic interface with other residues from the perpendicular helix (324-345) which maintains the rigidity of the transducer domain, an important feature for the transduction of the ATP hydrolysis signal.

5. Part of the CTD domain was assigned (1192-1215). Based on Supplementary Fig. 5g, there is a density break between the coiled coil region and the assigned stretch. This can make the residue assignment uncertain if there is a disordered loop of unknown length in between. Can the authors provide additional data/validation on this residue assignment?

We thank the reviewer for giving us the opportunity to clarify this point. The beginning of the CTD stretch is indeed not observable at a regular threshold in the EM map. We believe this part is highly flexible because there is no contact with the rest of the protein and it is mainly composed of glycine residues. However, when changing the threshold to lower RMSD in the unsharpened overall map (4.7 Å), the density becomes visible showing a clear connection from the coiled-coil domain to the CTD linker (see below). We have added the comparison in Supplemental Fig. 5.

Interestingly, in bacteria, the linker to the CTD also takes a shorter but similar path leading to the CTD. The difference is that in the human TopII alpha structure, the last coiled-coil helix is much longer forcing the linker to take longer route to the CTD stretching below the DNA G-segment.

6. Based on deletion mutants, this region is proposed to be a positive regulator of the relaxation activity. In the context of the FL construct, there is an autoinhibitory mechanism involved? If so, how would this be released to allow for the positive influence of this part of the CTD?

In the context of the full-length structure, we believe that the CTD bulk (38 kDa), probably entirely or partially unfolded, down-regulates the activity of the Human TopII. There is no clear answer to us on how this inhibition would be released entirely, except maybe by directed proteolysis which is unlikely to happen in the cell.

Nonetheless, the CTD is subjected to a host of post-translational modifications (at least 30) which is thought to modulate the TopII activities (doi: 10.1038/s41598-018-27606-8). Particularly in the case of the CTD linker, T1205 and S1213 are found to be phosphorylated in cancer cells (PhosphoSitePlus and doi: 10.20517/cdr.2019.114). The modification of these residues could affect positively or negatively the binding of the CTD linker to the DNA-binding/cleavage domain and hence modulate the activity of the Topo IIalpha.

Minor points:

7. The term “near-atomic resolution” is debatable these days. Many have argued that atomic would be 1Å, near-atomic could actually be 1-2 Å rather than 4-5 Å we see in EM (See Wlodawer & Dauter Acta Cryst D. 2017 for more details) (Line 89).

We thank the reviewer for this comment, also raised by reviewer #2, and for providing the reference. As mentioned earlier in our response to the reviewers, we are aware that the EM field has been evolving extremely fast in the recent years and what was considered “near-atomic resolution” three years ago (3-4 Å) is now medium resolution. We modified this statement in the text whenever applicable.

8. The EM density fit shown in Fig. 1d and Supp. Fig. 5f for etoposide is not very clear as stated in the manuscript. Any comments on this? Perhaps a better view or better of showing this will be more convincing.

Thanks to the reviewer's request for reprocessing the all the EM data, we have improved the resolution of the map DNA-binding/cleavage domain map from 4.1 Å to 3.6 Å. By doing so, the density around the etoposide molecule has dramatically improved (3 Å local resolution) and can now be clearly identified. We have taken into account the reviewer's remark and have updated our Figure 1 and Supplementary Fig. 5 and hope that the views we have chosen are now more convincing.

9. Line 111-122, the authors discussed the cavity to accommodate T-segment on top of G-segment, C-gate, N-gate etc and citing Fig. 3. Fig. 3 could be improved to label these features clearly for clarity. For an audience not familiar the topoisomerase field, it is very difficult to follow these terms without an illustration. This is applied for the remainders of the figures and texts.

We agree with the reviewer that these terms are very specific to the topoisomerase field and we aim to be of interest for a wide readership. We have added a schematic in Figure 3a showing the different modules (gates) of the Topo II, as well as the different DNA segments cleaved and travelling through the enzyme during the catalytic cycle. We have also labelled the different modules in each panel to show the movement between the two states for each module. We have modified the main text accordingly to direct the reader to the schematic when these particular terms are used.

10. Fig. 4e, F427S is a mistake?

We thank the reviewer for finding this mistake. We meant F₄₁₇S. This has been corrected.

11. Line 221, prokaryotic

We thank the reviewer for finding this typographical error. We have corrected this.

Reviewers' Comments:

Reviewer #1:

Remarks to the Author:

The manuscript by Lamour et al. describes the cryo-EM structure of the complete human topoisomerase II alpha complex with DNA. It is the first cryo-EM structure of the complete enzyme and only the second structure of a kinetically competent eukaryotic type II enzyme (the first was the crystal structure of yeast topoisomerase II that lacked only its C-terminal domain). The work is timely and addresses important issues in the field. The authors provide structural evidence for a novel regulatory element in human topoisomerase II alpha that links the DNA binding/cleavage domain and the C-terminal domain. Overall, the work by the Lamour laboratory is nicely done, and makes an important contribution to the topoisomerase field. The manuscript should definitely be of interest to readers of Nature Communications.

The revised manuscript is substantially improved and adequately addresses all of the issues that I raised during the initial review process. I believe that the conclusions of the work are justified.

Reviewer #2:

Remarks to the Author:

The authors have answered well most of the comments from the reviewers. The manuscript is stronger and better now. The structures are at higher resolution and the biochemical data is also stronger.

Reviewer #3:

Remarks to the Author:

The authors have done a great job at addressing comments from all three reviewers. I am satisfied with the revised manuscript. The improved maps and additional biochemical experiments performed here made the conclusions drawn from the paper very solid. This paper will be of great interest to Nature Communications readers.